# On the explainable properties of 1-Lipschitz Neural Networks: An Optimal Transport Perspective

**Mathieu Serrurier**[1]  **Franck Mamalet**[2]  **Thomas Fel**[3,4]  **Louis Béthune**[1]

**Thibaut Boissin**[2]

[1]Université Paul-Sabatier, IRIT, Toulouse, France
[2]Institut de Recherche Technologique Saint-Exupéry, Toulouse, France
[3]Carney Institute for Brain Science,Brown University, USA
[4]Innovation & Research Division, SNCF, France

## Abstract

Input gradients have a pivotal role in a variety of applications, including adversarial attack algorithms for evaluating model robustness, explainable AI techniques for generating Saliency Maps, and counterfactual explanations. However, Saliency Maps generated by traditional neural networks are often noisy and provide limited insights. In this paper, we demonstrate that, on the contrary, the Saliency Maps of 1-Lipschitz neural networks, learned with the dual loss of an optimal transportation problem, exhibit desirable XAI properties: They are highly concentrated on the essential parts of the image with low noise, significantly outperforming state-of-the-art explanation approaches across various models and metrics. We also prove that these maps align unprecedentedly well with human explanations on ImageNet. To explain the particularly beneficial properties of the Saliency Map for such models, we prove this gradient encodes both the direction of the transportation plan and the direction towards the nearest adversarial attack. Following the gradient down to the decision boundary is no longer considered an adversarial attack, but rather a counterfactual explanation that explicitly transports the input from one class to another. Thus, Learning with such a loss jointly optimizes the classification objective and the alignment of the gradient, i.e. the Saliency Map, to the transportation plan direction. These networks were previously known to be certifiably robust by design, and we demonstrate that they scale well for large problems and models, and are tailored for explainability using a fast and straightforward method.

## 1  Introduction

The Lipschitz constant of a function expresses the extent to which the output may vary for a small shift in the input. As a composition of numerous functions, the Lipschitz constant of a neural network can be arbitrarily high, particularly when trained for a classification task [8]. Adversarial attacks [45] exploit this weakness by selecting minor modifications, such as imperceptible noise, for a given example to change the predicted class and deceive the network. Consequently, Saliency Maps [59] – gradient of output with respect to the input –, serve as the basis for most adversarial attacks and often highlight noisy patterns that fool the model instead of meaningful modifications, rendering them generally unsuitable for explaining model decisions. Therefore, several methods requiring more complex computations, such as SmoothGrad [61], Integrated Gradient [64], or Grad-CAM [55], have been proposed to provide smoother explanations. Recently, the XAI community has investigated the link between explainability and robustness and proposed methods and metrics

37th Conference on Neural Information Processing Systems (NeurIPS 2023).

accordingly [35, 12, 42, 53]. However, the reliability of those automatic metrics can be compromised by artifacts introduced by the baselines [35, 63, 31, 38, 30], and there is no conclusive evidence demonstrating their correlation with the human understanding of explanations. To address this, a study by [22] suggests completing those metrics with the alignment between attribution methods and human feature importance using the ClickMe dataset [43].

In [56], authors propose to address the weakness with respect to adversarial attacks by training 1-Lipschitz constrained neural networks with a loss that is the dual of an optimal transport optimization problem, called hKR (Eq. 1). The resulting models have been shown to be robust with a certifiable margin. We refer to these networks as Optimal Transport Neural Networks (OTNN) hereafter.

In this paper, we demonstrate that OTNNs exhibit valuable explainability properties. Our experiments reveal that OTNN Saliency Maps significantly outperform various attribution methods for unconstrained networks across all tested state-of-the-art Explainable XAI metrics. This improvement is consistent across toy datasets, large image datasets, binary, and multiclass problems. Qualitatively, OTNN Saliency Maps concentrate on crucial image regions and generate less noise than maps of unconstrained networks, as illustrated in Figure 1. Figure 1.c presents the Saliency Maps of an OTNN based on ResNet50 alongside its unconstrained vanilla counterpart, both trained on ImageNet [16]. In the unconstrained case, the Saliency Maps appear sparse and uninformative, with the most critical pixels often located outside the subject. Conversely, the OTNN Saliency Map is less noisy and highlights significant image features. This distinction is emphasized in Figure 1.d), comparing the feature visualization of the two models. Feature visualization extracts the inverse prototypical image for a given class using gradient ascent [48, 47]. The results for vanilla ResNet are noticeably noisy, making class identification difficult. In contrast, feature visualization with OTNN yields clearer results, displaying easily identifiable features and classes (e.g., goldfish, butterfly, and medusa). Furthermore, modifying the image following the gradient direction provides an interpretable method for altering the image to change its class. Pictures 1.a) and 1.b) display the original image, the gradient direction, and the transformation following the gradient direction (refer to Section 4 for details). We observe explicit structural changes in the images, transforming them into an example of another class in both multiclass (MNIST) and high-resolution multi-labels cases (e.g., smile/not smile and blond hair/not blond hair classification). Lastly, a large-scale human experiment demonstrates that these maps are remarkably aligned with human attribution on ImageNet (Fig. 4).

We provide a theoretical justification for the well-behaved OTNN Saliency Maps. Building upon the fact that OTNNs encode the dual formulation of the optimal transport problem, we prove that the gradient of the optimal solution at a given point $x$ is both (i) in the direction of the nearest adversarial example on the decision boundary, and (ii) in the direction of the image of $x$ according to the underlying transport plan. This implies that adversarial attacks for an OTNN are equivalent to traversing the optimal transport path which can be achieved by following the gradient. Consequently, the resulting modification serves both as an adversarial attack and a counterfactual explanation, explaining why the decision was A and not B [40]. An optimal transport plan between two classes can be interpreted as a global approach for constructing counterfactuals, as suggested in [11, 15]. These counterfactuals may not correspond to the smallest transformation for a given input sample but rather the smallest transformation on average when pairing points from two classes. A consequence of this property is that the Saliency Map of an OTNN for an image indicates the importance of each pixel in the modifications needed to change the class. It is worth noting that several methods based on GAN [36] or causality penalty [37] produce highly realistic counterfactual images. However, the objective of our paper is not to compete with the quality of these results, but rather to demonstrate that OTNN Saliency Maps possess both theoretical and empirical foundations as counterfactual explanations.

We summarize our contributions as follows: first, after introducing the background on OTNN and XAI, we establish several properties of the gradient of an OTNN with respect to adversarial attacks, decision boundaries, and optimal transport. Second, we establish that the optimal transport properties of OTNN's gradient lead to a reinterpretation of adversarial attacks as counterfactual explanations, consequently endowing the Saliency Map with the favorable XAI properties inherent in these explanations. Third, our experiments support the theoretical results, showing that metric scores are higher for most of the XAI methods on OTNN compared to unconstrained neural networks. Additionally, we find that the Saliency Map for OTNN achieves top-ranked scores on XAI metrics compared to more sophisticated XAI methods, and is equivalent to Smoothgrad. Lastly, drawing from [22], we emphasize that OTNNs are naturally and remarkably aligned with human explanations, and we present several examples of gradient-based counterfactuals obtained with OTNNs.

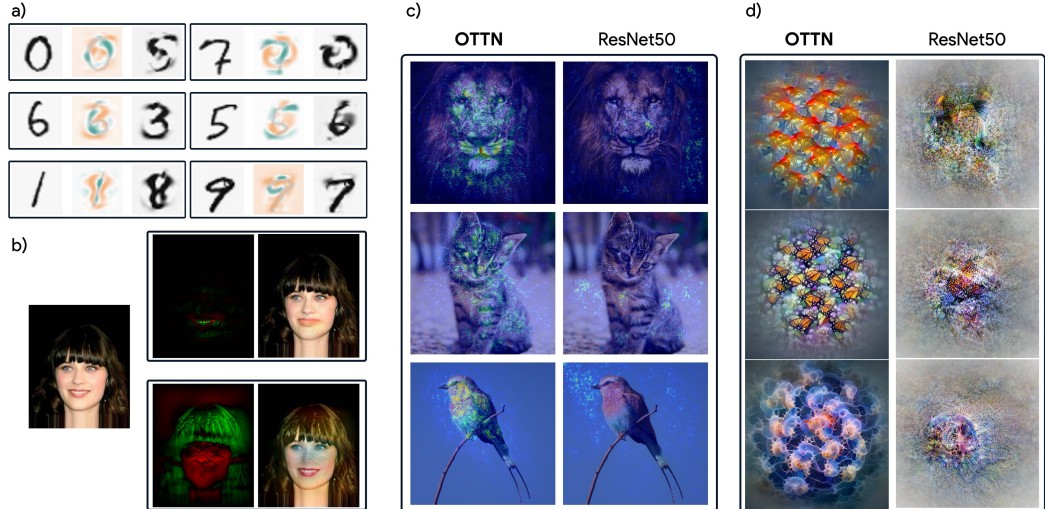

Figure 1: **Illustration of the beneficial properties of OTNN gradients.** Examples **a)** and **b)** show that the gradients naturally provide a direction that enables the generation of adversarial images - a theoretical justification based on optimal transport is provided in Section 3. By applying the gradient $\boldsymbol{x}' = \boldsymbol{x} - t\nabla_{\boldsymbol{x}}\boldsymbol{f}(\boldsymbol{x})$ to the original image $\boldsymbol{x}$ (on the left), any digit from MNIST can be transformed into its counterfactual $\boldsymbol{x}'$ (e.g., turning a 0 into a 5). In **b)**, we illustrate that this approach can be applied to larger datasets, such as Celeb-A, by creating two counterfactual examples for the closed-mouth and blonde classes. In **c)**, we compare the Saliency Map of a classical model with those of OTNN gradients, which are more focused on relevant elements. Finally, in **d)**, we show that following the gradients of OTNN could generate convincing feature visualizations that ease the understanding of the model's features.

## 2 Related work

**1-Lipschitz Neural network and optimal transport:** Consider a classical supervised machine learning binary classification problem on $(\Omega, \mathcal{F}, \mathbb{P})$ – the underlying probability space – where $\Omega$ is the sample space, $\mathcal{F}$ is a $\sigma$-algebra on $\Omega$, and $\mathbb{P}$ is a probability measure on $\mathcal{F}$. We denote the input space $\mathcal{X} \subseteq \mathbb{R}^d$ and the output space $\mathcal{Y} = \{\pm 1\}$. Let input data $\boldsymbol{x} : \Omega \to \mathcal{X}$ and target label $y : \Omega \to \mathcal{Y}$ are random variables with distributions $P_{\boldsymbol{x}}, P_y$, respectively. The joint random vector $(\boldsymbol{x}, y)$ on $(\Omega, \mathcal{F})$ has a joint distribution $P$ defined over the product space $\mathcal{X} \times \mathcal{Y}$. Moreover, let $\mu = P(\boldsymbol{x}|y = 1)$ and $\nu = P(\boldsymbol{x}|y = -1)$ the conditional probability distributions of $\boldsymbol{x}$ given the true label. We assume that the supports of $\Omega$, $\mu$ and $\nu$ are compact sets.

A function $\boldsymbol{f} : \mathcal{X} \to \mathbb{R}$ is a 1-Lipschitz functions over $\mathcal{X}$ (denoted $Lip_1(\mathcal{X})$) if and only if $\forall (\boldsymbol{x}, \boldsymbol{z}) \in \mathcal{X}^2, ||\boldsymbol{f}(\boldsymbol{x}) - \boldsymbol{f}(\boldsymbol{z})|| \leq ||\boldsymbol{x} - \boldsymbol{z}||$. 1-Lipschitz neural networks have received a lot of attention, especially due to the link with adversarial attacks. They provide certifiable robustness guarantees [32, 49], improve the generalizations [62] and the interpretability of the model [67]. The simplest way to constrain a network to be in $Lip_1(\mathcal{X})$ is to impose this 1-Lipschitz property to each layer. Frobenius normalization [54], or spectral normalization [46] can be used for linear layers, and can also be extended, in some situations, to orthogonalization [41, 1, 66, 6].

Optimal transport, 1-Lipschitz neural networks, and binary classification were first associated in the context of Wasserstein GAN (WGAN) [7]. The discriminator of a WGAN is the solution to the Kantorovich-Rubinstein dual formulation of the 1-Wasserstein distance [69], and it can be regarded as a binary classifier with a carefully chosen threshold. Nevertheless, it has been demonstrated in [56] that this type of classifier is suboptimal, even on a toy dataset. In the same paper, the authors address the suboptimality of the Wasserstein classifier by introducing the hKR loss $\mathcal{L}^{hKR}$, which adds a hinge regularization term to the Kantorovich-Rubinstein optimization objective :

$$\mathcal{L}_{\lambda,m}^{hKR}(\boldsymbol{f}) = \mathop{\mathbb{E}}_{\boldsymbol{x} \sim \nu}[\boldsymbol{f}(\boldsymbol{x})] - \mathop{\mathbb{E}}_{\boldsymbol{x} \sim \mu}[\boldsymbol{f}(\boldsymbol{x})] + \lambda \mathop{\mathbb{E}}_{(\boldsymbol{x},y) \sim P}(m - y\boldsymbol{f}(\boldsymbol{x}))_+ \tag{1}$$

where $m > 0$ is the margin, and $(z)_+ = max(0, z)$. We note $\boldsymbol{f}^*$ the optimal minimizer of $\mathcal{L}^{hKR}_{\lambda,m}$. The classification is given by the sign of $\boldsymbol{f}^*$. In the following, the 1-Lipschitz neural networks that minimize $\mathcal{L}^{hKR}_{\lambda,m}$ will be denoted as OTNN. Given a function $\boldsymbol{f}$, a classifier based on $\mathrm{sign}(\boldsymbol{f})$ and an example $\boldsymbol{x}$, an adversarial example is defined as follows:

$$adv(\boldsymbol{f}, \boldsymbol{x}) = \operatorname*{arg\,min}_{\boldsymbol{z} \in \mathcal{X}} \parallel \boldsymbol{x} - \boldsymbol{z} \parallel \quad s.t. \quad \mathrm{sign}(\boldsymbol{f}(\boldsymbol{z})) \neq \mathrm{sign}(\boldsymbol{f}(\boldsymbol{x})). \tag{2}$$

Since $\boldsymbol{f}^*$ is a 1-Lipschitz function, $|\boldsymbol{f}^*(\boldsymbol{x})|$ is a certifiable lower bound of the robustness of the classification of $\boldsymbol{x}$ (i.e. $\forall \boldsymbol{x} \in \mathcal{X}, |\boldsymbol{f}^*(\boldsymbol{x})| \leq ||\boldsymbol{x} - adv(\boldsymbol{f}^*, \boldsymbol{x})||$). The function $\boldsymbol{f}^*$ has the following properties [56] (*i*) if the supports of $\mu$ and $\nu$ are disjoints (separable classes) with a minimal distance of $\epsilon > 0$, then for $m < 2\epsilon$, $f^*$ achieves 100% accuracy; (*ii*) minimizing $\mathcal{L}^{hKR}$ is still the dual formulation of an optimal transport problem (see appendix for more details).

**Explainability and metrics:** Attribution methods aim to explain the prediction of a deep neural network by pointing out input variables that support the prediction – typically pixels or image regions for images – which lead to importance maps. Saliency [59] was the first proposed white-box attribution method and consists of back-propagating the gradient from the output to the input. The resulting absolute gradient heatmap indicates which pixels affect the most the decision score. However, this family of methods suffers from problems inherent to the gradients of standard models. Methods such as Integrated Gradient [64] and SmoothGrad [61] partially address this issue by accumulating gradients, either along a straight interpolation path from a baseline state to the original image or from a set of points close to the original image obtained after adding noise but multiply the computational cost by several orders of magnitude. These methods were then followed by a plethora of other methods using gradients such as Grad-cam [55] or Input Gradient [4]. All rely on the gradient calculation of the classification score. Finally, other methods – sometimes called black-box attribution methods – do not involve the gradient and rely on perturbations around the image to generate their explanations [50, 20].

However, it is becoming increasingly clear that current methods raise many issues [2, 33, 60] such as confirmation bias: it is not because the explanations make sense to humans that they reflect the evidence of the prediction. To address this challenge, a large number of metrics were proposed to provide objective evaluations of the quality of explanations. Deletion and Insertion methods [50] evaluate the drop in accuracy when important pixels are replaced by a baseline. $\mu$Fidelity method [9] evaluates the correlation between the sum of importance scores of pixels and the drop of the score when removing these pixels. In parallel, a growing literature relies on model robustness to derive new desiderata for a good explanation [35, 12, 42, 53, 21]. In addition, [35] showed that some of these metrics also suffer from a bias due to the choice of the baseline value and proposed a new metric called Robustness-Sr. This metric assesses the ease to generate an adversarial example when the attack is limited to the important variables proposed by the explanation. Other metrics consider properties such as generalizability, consistency [23], or stability [74, 9] of explanation methods. A recent approach [43] aims to evaluate the alignment between attribution methods and human feature importance across 200,000 unique ImageNet images (called ClickMe dataset). The alignment between DNN Saliency and human explanations is quantified using the mean Spearman correlation, normalized by the average inter-rater alignment of humans.

These works on explainability metrics have also initiated the emergence of links between the robustness of models and the quality of their explanations [14, 72, 57, 58, 18, 19]. In particular,[23] claimed that 1-Lipschitz networks explanations have better metrics scores. But this study was not on OTNNs and was limited to their proposed metrics.

To end with, recent literature is focusing on counterfactual explanations [70, 68] methods, providing information on "why the decision was A and not B". Several properties are desirable for these counterfactual explanations[68]: Validity (close sample and in another class), Actionability, Sparsity, Data Manifold closeness, and Causality. The three last properties are generally not covered by standard adversarial attacks and complex methods have been proposed [28, 52, 71]. Since often a causal model is hard to fully-define, recent papers [11, 15] have proposed a definition of counterfactual based on optimal transport easier to compute and that can sometimes coincide with causal model based ones. We will rely on this theoretical definition of counterfactuals.

# 3   Theoretical properties of OTNN gradient

In this section, we extend the properties of the OTNNs to the explainability framework, all the proofs are in the appendix A.1. We note $\pi$ the optimal transport plan corresponding to the minimizer of $\mathcal{L}_{\lambda,m}^{hKR}$. In the most general setting, $\pi$ is a joint distribution over $\mu, \nu$ pairs. However when $\mu$ and $\nu$ admit a density function [51] with respect to Lebesgue measure, then the joint density describes a deterministic mapping, i.e. a Monge map. Given $\boldsymbol{x} \sim \mu$ (resp. $\nu$) we note $\boldsymbol{z} = \boldsymbol{\gamma}_\pi(\boldsymbol{x}) \in \nu$ (resp. $\mu$) the image of $\boldsymbol{x}$ with respect to $\pi$. When $\pi$ is not deterministic (on real datasets that are defined as a discrete collection of Diracs), we take $\boldsymbol{\gamma}_\pi(\boldsymbol{x})$ as the point of maximal mass with respect to $\pi$.

**Proposition 1 (Transportation plan direction)** *Let $\boldsymbol{f}^*$ an optimal solution minimizing the $\mathcal{L}_{\lambda,m}^{hKR}$. Given $\boldsymbol{x} \sim \mu$ (resp. $\nu$) and $\boldsymbol{z} = \boldsymbol{\gamma}_\pi(\boldsymbol{x})$, then $\exists t \geq 0$ (resp. $t \leq 0$) such that $\boldsymbol{\gamma}_\pi(\boldsymbol{x}) = \boldsymbol{x} - t\nabla_{\boldsymbol{x}}\boldsymbol{f}^*(\boldsymbol{x})$ almost surely.*

This proposition also holds for the Kantorovich-Rubinstein dual problem without hinge regularization, demonstrating that for $\boldsymbol{x} \sim P$, the gradient $\nabla_{\boldsymbol{x}}\boldsymbol{f}^*(\boldsymbol{x})$ indicates the direction in the transportation plan almost surely.

**Proposition 2 (Decision boundary)** *Let $\mu$ and $\nu$ two distributions with disjoint supports with minimal distance $\epsilon$ and $\boldsymbol{f}^*$ an optimal solution minimizing the $\mathcal{L}_{\lambda,m}^{hKR}$ with $m < 2\epsilon$. Given $\boldsymbol{x} \sim P$, $\boldsymbol{x}_\delta = \boldsymbol{x} - \boldsymbol{f}^*(\boldsymbol{x})\nabla_{\boldsymbol{x}}\boldsymbol{f}^*(\boldsymbol{x}) \in \partial\mathcal{X}$ where $\partial\mathcal{X} = \{\boldsymbol{x}' \in \mathcal{X} | \boldsymbol{f}^*(\boldsymbol{x}') = 0\}$ is the decision boundary (i.e. the 0 level set of $\boldsymbol{f}^*$).*

Experiments suggest this probably remains true when the supports of $\mu$ and $\nu$ are not disjoint. Prop. 2 proves that for an OTNN $\boldsymbol{f}$ learnt by minimizing the $\mathcal{L}_{\lambda,m}^{hKR}$, $|\boldsymbol{f}(\boldsymbol{x})|$ provides a tight robustness certificate. A direct consequence of 2, is that $t$ defined in 1 is such that $|t| \geq |\boldsymbol{f}^*(\boldsymbol{x})|$.

**Corollary 1** *Let $\mu$ and $\nu$ two separable distributions with minimal distance $\epsilon$ and $\boldsymbol{f}^*$ an optimal solution minimizing the $\mathcal{L}_{\lambda,m}^{hKR}$ with $m < 2\epsilon$, given $\boldsymbol{x} \sim P$, $adv(\boldsymbol{f}^*, \boldsymbol{x}) = \boldsymbol{x}_\delta$ almost surely, where $\boldsymbol{x}_\delta = \boldsymbol{x} - \boldsymbol{f}^*(\boldsymbol{x})\nabla_{\boldsymbol{x}}\boldsymbol{f}^*(\boldsymbol{x})$.*

This corollary shows that adversarial examples are precisely identified for the classifier based on $\mathcal{L}_{\lambda,m}^{hKR}$: the direction given by $\nabla_{\boldsymbol{x}}\boldsymbol{f}^*(\boldsymbol{x})$ and distance by $|\boldsymbol{f}^*(\boldsymbol{x})| * ||\nabla_{\boldsymbol{x}}\boldsymbol{f}^*(\boldsymbol{x})|| = |\boldsymbol{f}^*(\boldsymbol{x})|$. In this scenario, the optimal adversarial attacks align with the gradient direction (i.e., FGSM attack [27]). This supports the observations made in [56], where all attacks, such as PGD [45] or Carlini and Wagner [13], applied on an OTNN model, were equivalent to FGSM attacks.

To illustrate these propositions, we learnt a dense binary classifier with $\mathcal{L}_{\lambda,m}^{hKR}$ to separate two complex distributions, following two concentric Koch snowflakes. Fig.2-a shows the two distributions (blue and orange snowflakes), the learnt boundary (0-levelset) (red dashed line). Fig.2-b,c show for random samples $\boldsymbol{x}$ from the two distributions, the segments $[\boldsymbol{x}, \boldsymbol{x}_\delta]$ where $\boldsymbol{x}_\delta$ is defined in Prop. 2 . As expected by Prop. 2, $\boldsymbol{x}_\delta$ points fall exactly on the decision boundary. Besides, as stated in Prop. 1 each segment provides the direction of the image with respect to the transport plan.

Finally, we showed that with OTNN, adversarial attacks are formally known and simple to compute. Furthermore, since we proved that these attacks are along the transportation map, they acquire a meaningful interpretation and are no longer mere adversarial examples exploiting local noise, but rather correspond to the global solution of a transportation problem.

# 4   Link between OTNN gradient and counterfactual explanations

The vulnerability of traditional networks to adversarial attacks indicates that their decision boundaries are locally flawed, deviating significantly from the Bayesian boundaries between classes. Since the gradient directs towards this anomalous boundary, Saliency Maps [59], given by $g(\boldsymbol{x}) = |\nabla_{\boldsymbol{x}}\boldsymbol{f}(\boldsymbol{x})|$ fails to represent a meaningful transition between classes and then often lead to noisy explanation (as stated in Section 2).

On the contrary, in the experiments, we will demonstrate that OTNN gradients induce meaningful explanations (Sec. 5). We justify these good properties by building a link with counterfactual

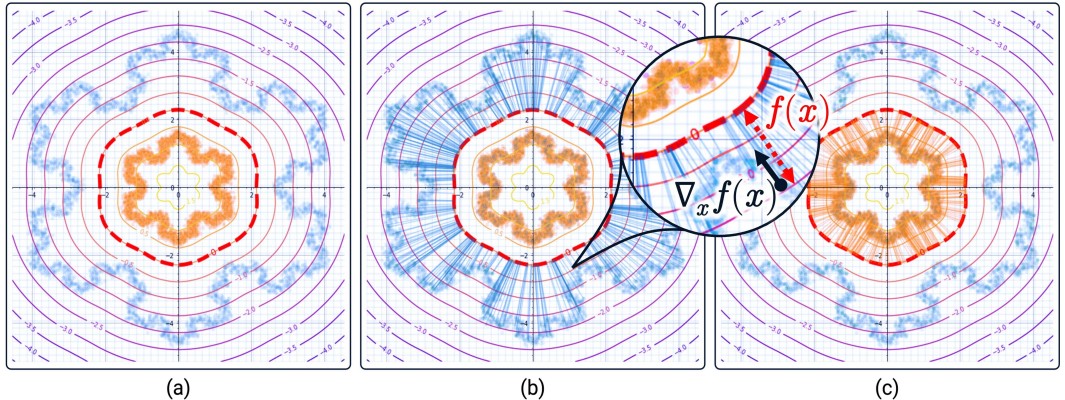

Figure 2: Level sets of an OTNN classifier $\boldsymbol{f}$ for two concentric Koch snowflakes (a). The decision boundary (denoted $\partial\mathcal{X}$, also called the 0-level set) is the red dashed line. Figure (b) (resp. (c)) represents the translation of the form $\boldsymbol{x}' = \boldsymbol{x} - \boldsymbol{f}(\boldsymbol{x})\nabla_{\boldsymbol{x}}\boldsymbol{f}(\boldsymbol{x})$ of each point $\boldsymbol{x}$ of the first class (resp second class). $[\boldsymbol{x}, \boldsymbol{x}']$ pairs are represented by blue (resp. orange) segments.

explanations. Indeed, according to [11, 15], optimal transport plans are potential surrogates or even substitutes to causal counterfactuals. Optimal Transport plan provides for any sample $\boldsymbol{x} \in \mu$ a sample $\boldsymbol{\gamma}_\pi(\boldsymbol{x}) \in \nu$, the closest in average on the pairing process. [15] prove that these optimal transport plans can even coincide with causal counterfactuals when available. Relying on this definition of OT counterfactual, Prop. 1 demonstrates that gradients of the optimal OTNN solution provide almost surely the direction to the counterfactual $\boldsymbol{\gamma}_\pi(\boldsymbol{x}) = \boldsymbol{x} - t\nabla_{\boldsymbol{x}}\boldsymbol{f}^*(\boldsymbol{x})$. Even if $t$ is only partially known, using $t = \boldsymbol{f}^*(\boldsymbol{x})$, we know that $\boldsymbol{x}_\delta$ is on the decision boundary (Corr. 1) and is both an adversarial attack and a counterfactual explanation and $|t| \geq |\boldsymbol{f}^*(\boldsymbol{x})|$ is on the path to the other distribution.

Thus the learning process of OTNNs induces a strong constraint on the gradients of the neural network, aligning them to the optimal transport plan. We claim that is the reason why the simple Saliency Maps for OTNNs have very good properties: We will demonstrate in the Sec 5 that, for the Saliency Map explanations: (***i***) metrics scores are higher or comparable to other explanation methods (which is not the case for unconstrained networks), thus it has higher ranks; (***ii***) distance to other attribution methods such as Smoothgrad is imperceptible; (***iii***) scores obtained on metrics that can be compared between networks are higher than those obtained with unconstrained networks; (***iv***) alignment with human explanations is impressive.

## 5  Experiments

We conduct experiments with networks learnt on FashionMNIST [73], and 22 binary labels of CelebA [44] datasets, Cat vs Dog (binary classification, 224x224x3 uncentered images), and Imagenet [17]. Note that labels in CelebA are very unbalanced (see Table 2 in Appendix A.2, with for instance less than 5% samples for *Mustache* or *Wearing_Hat*).

Architectures used for OTNNs and unconstrained networks are similar (same number of layers and neurons, a VGG for FashionMNIST and CelebA, a ResNet50 for Cat vs Dog and Imagenet). We also train an alternative of ResNet50 OTNN with twice the number of parameters (50 M). Unconstrained networks use batchnorm and ReLU layers for activation, whereas OTNNs only use GroupSort2 [5, 56] activation. OTNNs are built using the *DEEL.LIP*[1] library, using Björck orthogonalization projection algorithm for linear layers. Note that several other approaches can be used for orthogonalization without altering the theoretical results; these might potentially enhance experimental outcome scores. The loss functions are cross-entropy for unconstrained networks (categorical for multiclass, and sigmoid for multilabel settings), and hKR $\mathcal{L}_{\lambda,m}^{hKR}$ (and the multiclass variants see appendix A.2.3) for OTNNs. We train all networks with Adam optimizer [39]. Details on architectures and parameters are given in Appendix A.2.

---

[1]`https://github.com/deel-ai/deel-lip` distributed under MIT License (MIT)

| | Dataset | | | | | | | |
|---|---|---|---|---|---|---|---|---|
| | Fash. MNIST | | CelebA | | Cat vs Dog | | Imagenet | |
| | OTNN | Uncst. | OTNN | Uncst. | OTNN | Uncst. | OTNN | Uncst. |
| Attribution | μ**Fidelity-Uniform** (↑ is better) | | | | | | | |
| Saliency | **0.156** | -0.001 | **0.244** | 0.052 | **0.091** | 0.080 | **0.240** | 0.004 |
| SmoothGrad | **0.114** | -0.001 | **0.248** | 0.018 | **0.012** | -0.004 | **0.001** | -0.002 |
| Integ. Grad. | **-0.005** | -0.013 | **0.149** | 0.093 | 0.022 | **0.024** | **0.046** | 0.022 |
| Grad. Input | -0.017 | **-0.009** | **0.168** | 0.074 | **0.013** | 0.009 | **0.009** | 0.000 |
| GradCam | **0.215** | 0.02 | **0.028** | 0.002 | **0.101** | 0.052 | 0.029 | **0.046** |
| Attribution | μ**Fidelity-Zero** (↑ is better) | | | | | | | |
| Saliency | **0.246** | 0.034 | **0.325** | 0.082 | **0.121** | 0.079 | **0.147** | 0.049 |
| SmoothGrad | **0.332** | 0.052 | **0.324** | 0.091 | **0.011** | -0.004 | 0.001 | **0.002** |
| Integ. Grad. | **0.543** | 0.134 | **0.400** | 0.125 | **0.037** | 0.027 | **0.057** | 0.023 |
| Grad. Input | **0.479** | 0.079 | **0.439** | 0.093 | **0.019** | 0.004 | **0.020** | -0.001 |
| GradCam | **0.161** | 0.046 | **0.127** | 0.061 | **0.136** | 0.049 | 0.048 | **0.068** |
| Attribution | **Stability Spearman rank** (↓ is better) | | | | | | | |
| Saliency | **0.59** | 0.91 | **0.51** | 0.77 | **0.58** | 0.69 | **0.60** | 0.74 |
| SmoothGrad | **0.55** | 0.82 | **0.52** | 0.95 | **0.64** | 0.82 | **0.62** | 0.82 |
| Integ. Grad. | **0.61** | 0.79 | **0.52** | 0.87 | **0.61** | 0.76 | **0.60** | 0.74 |
| **Distance Saliency smoothgrad** (↓ is better) | | | | | | | | |
| Saliency | **7.5e-03** | 7.0e-02 | **3.1e-4** | 1.4e-1 | **3.7e-8** | 4.1e-8 | **3.7e-8** | 4.3e-8 |

Table 1: Comparison of XAI metrics for different attributions methods and dataset for OTNN and unconstrained networks.

**Classification performance:** OTNN models achieve comparable results to unconstrained ones, confirming claims of [8]: they reach 88.5% average accuracy on FashionMNIST (Table 6), and 81% (resp. 82%) average Sensitivity (resp. Specificity) over labels on CelebA (Table 6 in Appendix A.3). We use Sensitivity and Specificity for CelebA to take into consideration the unbalanced labels. OTNNs achieve 96% accuracy (98% for the unconstrained version) on Cat vs Dog and 67% (75% for the unconstrained version) on Imagenet. The ResNet50 OTNN with 50M parameters achieves 70% accuracy on Imagenet.

We present the results of quantitative evaluations of XAI metrics to compare the Saliency Map method with other explanation methods on OTNN, and more generally compare XAI explanations methods on these networks and their unconstrained counterparts. On CelebA, we only present the results for the label *Mustache*, but results for the other labels are similar. Parameters for explanation methods and metrics are given in Appendix A.4. We have chosen to present in Table 1 two SoTA XAI metrics that enable comparison between OTNNs and unconstrained networks. μ**Fidelity metric** [9] is a well-known method that measures the correlation between important variables defined by the explanation method and the model score decreases when these variables are reset to a baseline state (or replaced by uniform noise). Another important property for explanations is their stability for nearby samples. In [74], the authors proposed Stability metrics based on the $L_2$ distance. To better evaluate this stability and make it comparable for different models, we replace the $L_2$ distance by $1 - \rho$, $\rho$ being the Spearman rank correlation. Other model-dependent metrics are described in the Appendix. Results from the 50M parameter ResNet OTNN are included in the human alignment study (Fig 4) to illustrate that enhancing the model's complexity can bolster both the accuracy and alignment. The following observations can be drawn from Table 1:

**Saliency Map on OTNNs exhibit more fidelity and stability :** We confirm and amplify the results in [23]. Table. 1 clearly states that for most of the explanation methods, the μFidelity, zero or uniform, is significantly higher for OTNNs. And above all, Saliency Map score for OTNNs is always higher than any other attribution method score for unconstrained models. A similar observation holds for the Stability Spearman rank : OTNN scores are better whatever the attribution method.

**Saliency Map method on OTNNs is equivalent to other attribution methods:** We observe that the scores from the Saliency Maps and other methods are very similar for OTNN, with Saliency Maps consistently ranking among the top attribution methods. For the unconstrained case, Saliency Maps are occasionally outperformed by other attribution methods. Notably, for the ResNet archi- tecture, attribution methods other than Saliency Maps and GradCAM yield more erratic results for

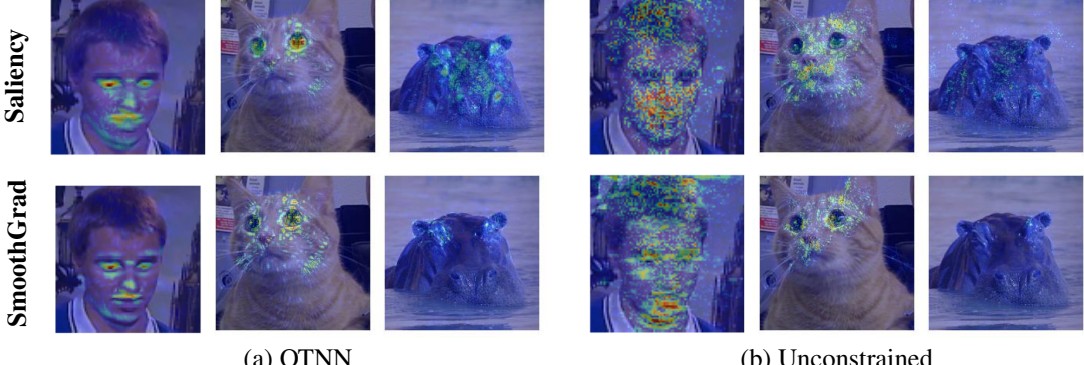

Figure 3: Comparison of Saliency Map and SmoothGrad explanations for (a) OTNN and (b) unconstrained network for, from left to right, CelebA, Cat vs Dog and Imagenet datasets.

fidelity metrics. To highlight these results, we compare the $L_2$ distance between Saliency Maps and SmoothGrad explanations, as suggested by [2, 23, 65, 26]. The explanation distances for OTNN are significantly lower than for the unconstrained ones and closely approach zero, indicating that for OTNN, averaging over a large set of noisy inputs—as in SmoothGrad—is unnecessary. This is illustrated in Fig.3.

**OTNNs explanations are aligned with human ones :** Adopting the method presented in [43], using the ClickMe dataset, we follow strictly their experimental methodology and use their code[2] to compute the human feature alignment of OTNN Saliency Maps and compare with the others models tested in [22]– more than 100 recent deep neural networks. In Figure 4, we demonstrate that OTNN models Saliency Maps, which also carries a theoretical interpretation as the direction of the transport plan, is more aligned with human attention than any other tested models and significantly surpasses the Pareto front discovered by [22]. The OTNN model is even more aligned than a ResNet50 model trained with a specific alignment objective, proposed by [22], and called *Harmonized ResNet50*. This finding is interesting as it indicates OTNNs are less prone to relying on spurious correlations [25] and better capture human visual strategies for object recognition. The implications of these results are crucial for both cognitive science and industrial applications. A model that more closely aligns with human attention and visual strategies can provide a more comprehensive understanding of how vision operates for humans, and also enhance the predictability, interpretability, and performance of object recognition models in industry settings. Furthermore, the drop in alignment observed in recent models highlights the necessity of considering the alignment of model visual strategies with human attention while developing object recognition models to reduce the reliance on spurious correlations and ensure that our models get things right for the right reasons.

**Qualitative results:** Using the learnt OTNN on FashionMNIST, CelebA (Mouth Slightly Open label), Cat vs Dog and Imagenet, Fig. 6,5 present the original image, average gradients $\nabla_x f_j$ over the channels, and images in the direction of the transport plan (Prop. 1), other samples are given in Appendix A.5. We can see that most of the gradients are visually consistent, showing clearly what has to be changed in the input image to modify the class and act as a counterfactual explanation. This is less clear for the Imagenet examples. This could be due to the difficulty of defining a transport plan for each pair of the 1000 classes. However, feature visualizations in Figure 1 show that the internal representation of the classes is still more interpretable than the unconstraint one. More generally, we observe that the gradient gives clear information about how the classifier makes its decision. For instance, for the cat, it shows that the classifier does not need to encode perfectly the concept of a cat, but mainly to identify the color of the eyes and size of the nose.

# 6 Conclusions and broader impact

In this paper, we study the explainability properties of OTNN (Optimal Transport Neural Networks) that are 1-Lipschitz constrained neural networks trained with an optimal transport dual loss. We

---

[2]https://github.com/serre-lab/Harmonization

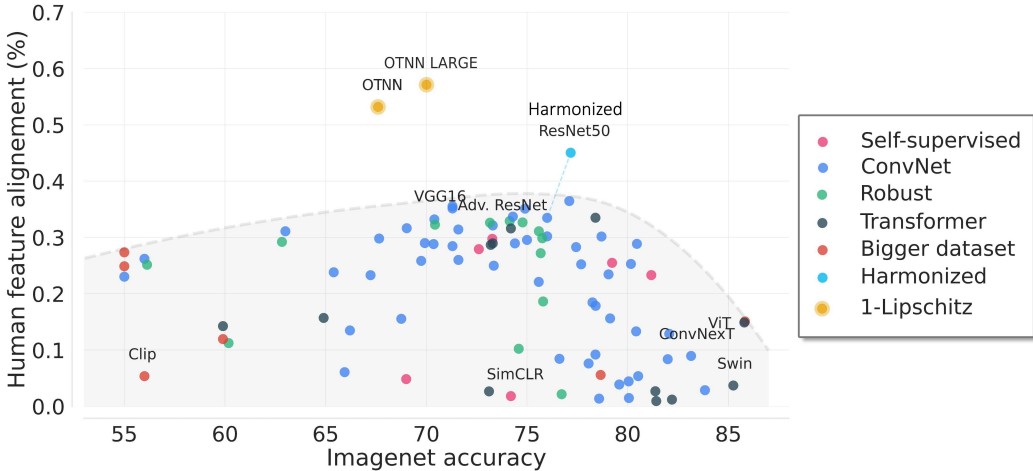

Figure 4: **OTNN are aligned with Human attention.** Our study shows that the Saliency Map of OTNN model is highly aligned with human attention.

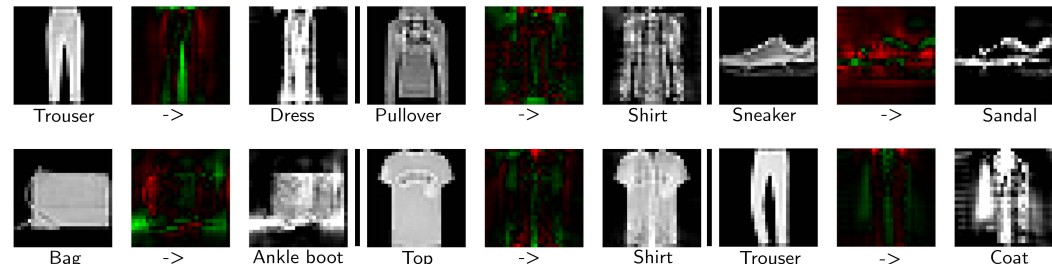

Figure 5: Samples of counterfactuals for FashionMNIST dataset on different classes and targets: (left) source image , (center) gradient image, (right) counterfactual of the form $\boldsymbol{x} - t * \hat{f}(\boldsymbol{x}) \nabla_x \hat{f}(\boldsymbol{x})$, for $t > 1$

establish that the gradient of the optimal solution aligns with the transportation plan direction, and the closest decision boundary point (adversarial example) also lies in this gradient direction at a distance of the absolute value of the network output. Relying on a formal definition of Optimal Transport counterfactuals, we build a link between the OTNN gradients and counterfactual explanations . We thus show that OTNNs loss jointly targets the classification task and induces a gradient alignment to the transportation plan. These beneficial properties of the gradient substantially enhance the Saliency Map XAI method for OTNN s. The experiments show that the simple Saliency Map has top-rank scores on state-of-the-art XAI metrics, and largely outperforms any method applied to unconstrained networks. Besides, as far as we know, our models are the first large provable 1-Lipschitz neural models that match state-of-the-art results on large problems. And even if counterfactual explanations are less compelling on Imagenet, probably due to the complexity of transport for large number of classes , we prove that OTNNs Saliency Maps are impressively aligned to human explanations.

**Broader impact.** This paper demonstrates the value of OTNNs for critical problems. OTNNs are certifiably robust and explainable with the simple Saliency Map method (highly aligned with human explanations) and have accuracy performances comparable to unconstrained networks.Though OTNNs take 3-6 times longer to train than unconstrained networks, they have similar computational costs during inference. We hope that this contribution will raise a great interest in these OTNN networks.

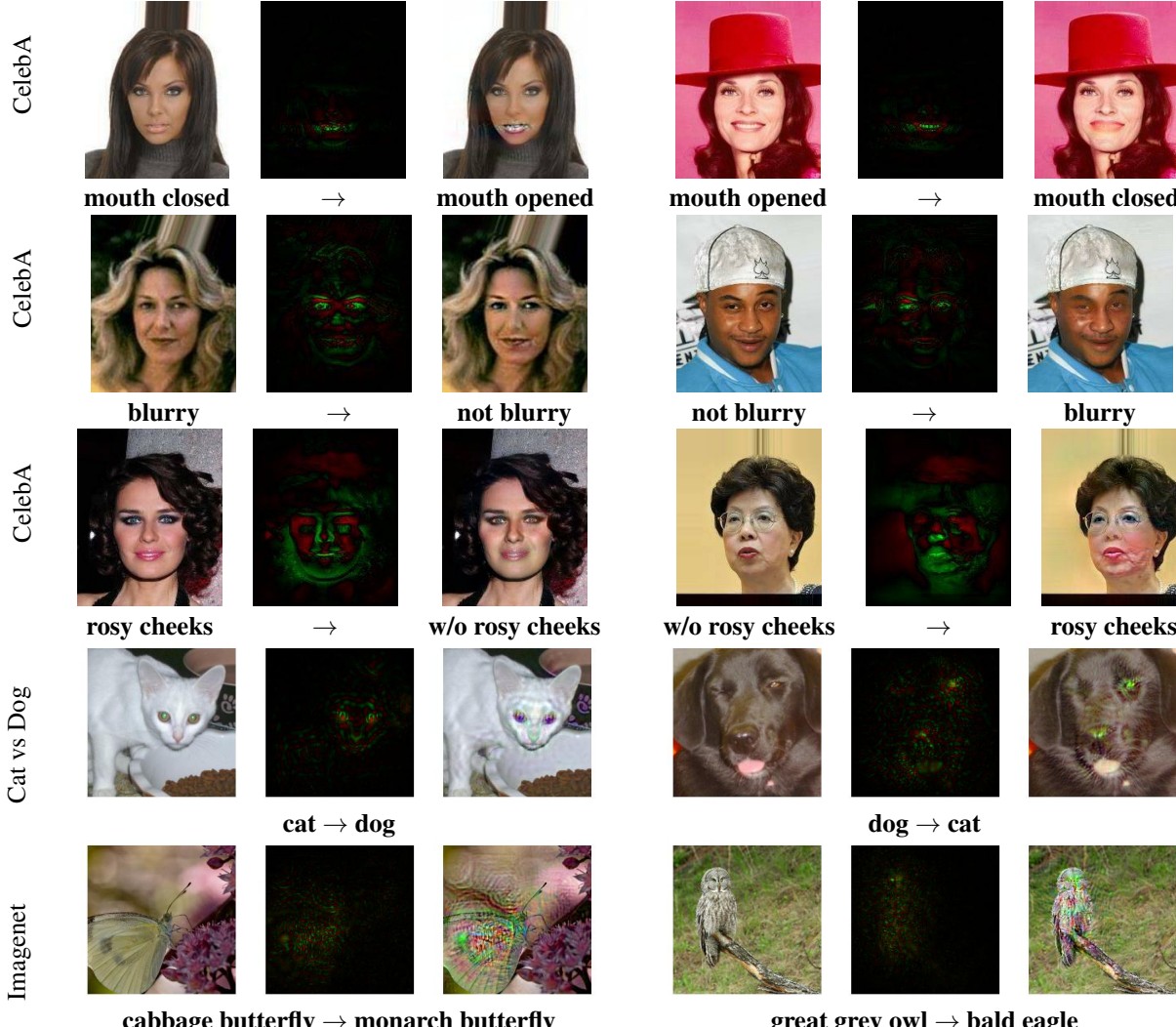

Figure 6: Samples of counterfactuals from different datasets: (left) source image , (center) gradient image, (right) counterfactual of the form $\boldsymbol{x} - t * \hat{f}(\boldsymbol{x})\nabla_x\hat{f}(\boldsymbol{x})$, for $t > 1$

## Acknowledgments and Disclosure of Funding

This work has benefited from the AI Interdisciplinary Institute ANITI, which is funded by the French "Investing for the Future – PIA3" program under the Grant agreement ANR-19-P3IA-0004. The authors gratefully acknowledge the support of the DEEL project.[3]

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

# A Appendix

## A.1 Additional definition and proofs

Let us first recall the optimal transport problem associated with the minimization of $\mathcal{L}^{hKR}_{\lambda,m}$:

$$\inf_{f\in Lip_1(\Omega)} \mathcal{L}^{hKR}_{\lambda(f),m} = \inf_{\pi\in\Pi^p_\lambda(\mu,\nu)} \int_{\Omega\times\Omega} |\boldsymbol{x}-\boldsymbol{z}|d\pi + \pi_{\boldsymbol{x}}(\Omega) + \pi_{\boldsymbol{z}}(\Omega) - 1 \tag{3}$$

Where $\Pi^p_\lambda(\mu,\nu)$ is the set consisting of positive measures $\pi\in\mathcal{M}_+(\Omega\times\Omega)$ which are absolutely continuous with respect to the joint measure $d\mu\times d\nu$ and $\frac{d\pi_{\boldsymbol{x}}}{d\mu}\in[p,p(m+\lambda)]$, $\frac{d\pi_{\boldsymbol{z}}}{d\nu}\in[1-p,(1-p)(m+\lambda)]$. We name $\pi^*$ the optimal transport plan according to Eq.3 and $f^*$ the associated potential function.

**Proof of proposition 1**: According to [56], we have

$$||\nabla_x f^*(\boldsymbol{x})|| = 1$$

almost surely and

$$\mathbb{P}_{(\boldsymbol{x},y)\sim\pi^*}\left(|f^*(\boldsymbol{x}) - f^*(y)| = ||\boldsymbol{x}-y||\right) = 1$$

Following the proof of proposition 1 in [29] and [3] we have :
Given $\boldsymbol{x}_\alpha = \alpha*x + (1-\alpha)y, 0\leq\alpha\leq 1$

$$\mathbb{P}_{(\boldsymbol{x},y)\sim\pi^*}\left(\nabla_x f^*(\boldsymbol{x}_\alpha) = \frac{\boldsymbol{x}_\alpha - y}{||\boldsymbol{x}_\alpha - y||}\right) = 1.$$

So for for $\alpha = 1$ whe have

$$\mathbb{P}_{(\boldsymbol{x},y)\sim\pi^*}\left(\nabla_x f^*(\boldsymbol{x}) = \frac{\boldsymbol{x} - y}{||\boldsymbol{x} - y||}\right) = 1$$

and then

$$\mathbb{P}_{(\boldsymbol{x},y)\sim\pi^*}\left(y = \boldsymbol{x} - \nabla_x f^*(\boldsymbol{x}).||\boldsymbol{x} - y||\right) = 1$$

This prove the proposition 1 by choosing $t = ||\boldsymbol{x} - y||$.∎

**Proof of proposition 2**: Let $\mu$ and $\nu$ two distributions with disjoint support with minimal distance $\epsilon$ and $f^*$ an optimal solution minimizing the $\mathcal{L}^{hKR}_{\lambda,m}$ with $m < 2\epsilon$. According to [56], $f^*$ is 100% accurate. Since the classification is based on the sign of f we have : $\forall\boldsymbol{x}\in\mu, f^*(\boldsymbol{x})\geq 0$ and $\forall y\in\nu, f^*(y)\leq 0$. Given $\boldsymbol{x}\in\mu$ and $y = tr_\pi(\boldsymbol{x}) = \boldsymbol{x} - t\nabla_x f^*(\boldsymbol{x})$ and $y\in\nu$. According to the previous proposition we have :

$$\begin{aligned}
|f^*(\boldsymbol{x}) - f^*(y)| &= ||\boldsymbol{x} - y|| \\
|f^*(\boldsymbol{x}) - f^*(y)| &= ||\boldsymbol{x} - (\boldsymbol{x} - t\nabla_x f^*(\boldsymbol{x}))|| \\
|f^*(\boldsymbol{x}) - f^*(y)| &= ||t\nabla_x f^*(\boldsymbol{x}))|| \\
|f^*(\boldsymbol{x}) - f^*(y)| &= t.||\nabla_x f^*(\boldsymbol{x}))|| & (t\geq 0) \\
|f^*(\boldsymbol{x}) - f^*(y)| &= t & (\nabla_x f^*(\boldsymbol{x}) = 1) \\
f^*(\boldsymbol{x}) - f^*(y) &= t & (f^*(\boldsymbol{x})\geq 0, f^*(y)\leq 0) \\
f^*(y) &= f^*(\boldsymbol{x}) - t
\end{aligned}$$

since $f^*(y)\leq 0$ we obtain :

$$f^*(\boldsymbol{x})\leq t$$

Since $f^*$ is continuous, $\exists t' > 0$ such that $\boldsymbol{x}_\delta = \boldsymbol{x} - t'\nabla_x f^*(\boldsymbol{x})$ and $f^*(\boldsymbol{x}_\delta) = 0$. We have :

$$\begin{aligned}
|f^*(\boldsymbol{x}) - f^*(\boldsymbol{x}_\delta)| &\leq ||\boldsymbol{x} - \boldsymbol{x}_\delta|| \\
f^*(\boldsymbol{x}) &\leq ||\boldsymbol{x} - (\boldsymbol{x} - t'\nabla_x f^*(\boldsymbol{x}))|| \\
f^*(\boldsymbol{x}) &\leq t'
\end{aligned}$$

and

$$|f^*(\boldsymbol{x}_\delta) - f^*(y)| \le ||\boldsymbol{x}_\delta - y||$$
$$-f^*(y) \le ||(\boldsymbol{x} - t'\nabla_x f^*(\boldsymbol{x})) - (\boldsymbol{x} - t\nabla_x f^*(\boldsymbol{x}))||$$
$$-f^*(y) \le t - t'$$
$$-f^*(y) \le ||\boldsymbol{x} - y|| - t' \qquad )$$

Then, if $f^*(\boldsymbol{x}) < t'$ we have

$$f^*(\boldsymbol{x}) - f^*(y) < t' + ||\boldsymbol{x} - y|| - t'$$
$$f^*(\boldsymbol{x}) - f^*(y) < ||\boldsymbol{x} - y||$$

which is a contradiction so $f^*(\boldsymbol{x}) = t'$ and

$$\boldsymbol{x}_\delta = \boldsymbol{x} - f^*(\boldsymbol{x})\nabla_x f^*(\boldsymbol{x})$$

∎

## A.2 Parameters and architectures

### A.2.1 Datasets

**FashionMNIST** has 50,000 images for training and 10,000 for test of size $28 \times 28 \times 1$, with 10 classes.

**CelebA** contains 162,770 training samples, 19,962 samples for test of size $218 \times 178 \times 3$. We have used a subset of 22 labels: *Attractive*, *Bald*, *Big_Nose*, *Black_Hair*, *Blond_Hair*, *Blurry*, *Brown_Hair*, *Eyeglasses*, *Gray_Hair*, *Heavy_Makeup*, *Male*, *Mouth_Slightly_Open*, *Mustache*, *Receding_Hairline*, *Rosy_Cheeks*, *Sideburns*, *Smiling*, *Wearing_Earrings*, *Wearing_Hat*, *Wearing_Lipstick*, *Wearing_Necktie*, *Young*.

Note that labels in CelebA are very unbalanced (see Table 2, with less than $5\%$ samples for *Mustache* or *Wearing_Hat* for instance). Thus we will use Sensibility and Specificity as metrics.

Table 2: CelebA label distribution: proportion of positive samples in training set (testing set) [bold: very unbalanced labels]

| *Attractive* | *Bald* | *Big_Nose* | *Black_Hair* | *Blond_Hair* |
|---|---|---|---|---|
| 0.51 (0.50) | **0.02 (0.02)** | **0.24 (0.21)** | **0.24 (0.27)** | **0.15 (0.13)** |
| *Blurry* | *Brown_Hair* | *Eyeglasses* | *Gray_Hair* | *Heavy_Makeup* |
| **0.05 (0.05)** | **0.20 (0.18)** | **0.06 (0.06)** | **0.04 (0.03)** | 0.38 (0.40) |
| *Male* | *Mouth_Slightly_Open* | *Mustache* | *Receding_Hairline* | *Rosy_Cheeks* |
| 0.42 (0.39) | 0.48 (0.50) | **0.04 (0.04)** | **0.08 (0.08)** | **0.06 (0.07)** |
| *Sideburns* | *Smiling* | *Wearing_Earrings* | *Wearing_Hat* | *Wearing_Lipstick* |
| **0.06 (0.05)** | 0.48 (0.50) | **0.19 (0.21)** | **0.05 (0.04)** | 0.47 (0.52) |
| *Wearing_Necktie* | *Young* | | | |
| **0.12 (0.14)** | 0.78 (0.76) | | | |

**Cat vs Dog** contains 17400 training samples, 5800 test samples of various size.

**Imagenet** contains 1M training samples, 100 000 samples for test of various size.

**preprocessing:** For FashionMNIST Images are normalized between $[0, 1]$ with no augmentation. For CelebA dataset, data augmentation is used with random crop, horizontal flip, random brightness, and random contrast. For imagenet and cat vs dog we use the standart preprocessing of resnet (with no normalization in $[0, 1]$)

### A.2.2 Architectures

As indicated in the paper, linear layers for OTNN and unconstrained networks are equivalent (same number of layers and neurons), but unconstrained networks use batchnorm and ReLU layer for activation, whereas OTNN only use GroupSort2 [5, 56] activation. OTNN are built using *DEEL.LIP*[4] library.

**1-Lipschitz networks parametrization.** Several solutions have been proposed to set the Lipschitz constant of affine layers: Weight clipping [7] (WGAN), Frobenius normalization [54] and spectral normalization [46]. In order to avoid vanishing gradients, orthogonalization can be done using Björck algorithm [10]. DEEL.LIP implements most of these solutions, but we focus on layers called *SpectralDense* and *SpectralConv2D*, with spectral normalization [46] and Björck algorithm [10]. Most activation functions are Lipschitz, including ReLU, sigmoid, but we use GroupSort2 proposed by [5], and defined by the following equation:

$$\text{GroupSort2}(x)_{2i,2i+1} = [\min(x_{2i}, x_{2i+1}), \max(x_{2i}, x_{2i+1})]$$

Network architectures used for CelebA dataset are described in Table 3.

Network architectures used for FashionMNIST dataset are described in Table 4. The same OTNN architecture is used for MNIST expermentation presented in Fig. 1.

---

[4]`https://github.com/deel-ai/deel-lip` distributed under MIT License (MIT)

Table 3: CelebA Neural network architectures: Sconv2D is SpectralConv2D, GS2 is GroupSort2, L2Pool is L2NormPooling, SDense is SpectralDense, BN is BatchNorm, AvgPool is AveragePooling

| Dataset | OTNN | Unconstrained NN | |
| --- | --- | --- | --- |
| | Layer | Layer | Output size |
| CelebA | Input | Input | $218 \times 178 \times 3$ |
| | SConv2D, GS2 | Conv2D, BN, ReLU | $218 \times 178 \times 16$ |
| | SConv2D, GS2 | Conv2D, BN, ReLU | $218 \times 178 \times 16$ |
| | L2Pool | AvgPool | $109 \times 89 \times 16$ |
| | SConv2D, GS2 | Conv2D, BN, ReLU | $109 \times 89 \times 32$ |
| | SConv2D, GS2 | Conv2D, BN, ReLU | $109 \times 89 \times 32$ |
| | L2Pool | AvgPool | $54 \times 44 \times 32$ |
| | SConv2D, GS2 | Conv2D, BN, ReLU | $54 \times 44 \times 64$ |
| | SConv2D, GS2 | Conv2D, BN, ReLU | $54 \times 44 \times 64$ |
| | SConv2D, GS2 | Conv2D, BN, ReLU | $54 \times 44 \times 64$ |
| | L2Pool | AvgPool | $27 \times 22 \times 64$ |
| | SConv2D, GS2 | Conv2D, BN, ReLU | $27 \times 22 \times 128$ |
| | SConv2D, GS2 | Conv2D, BN, ReLU | $27 \times 22 \times 128$ |
| | SConv2D, GS2 | Conv2D, BN, ReLU | $27 \times 22 \times 128$ |
| | L2Pool | AvgPool | $13 \times 11 \times 128$ |
| | SConv2D, GS2 | Conv2D, BN, ReLU | $13 \times 11 \times 128$ |
| | SConv2D, GS2 | Conv2D, BN, ReLU | $13 \times 11 \times 128$ |
| | SConv2D, GS2 | Conv2D, BN, ReLU | $13 \times 11 \times 128$ |
| | L2Pool | AvgPool | $6 \times 5 \times 128$ |
| | Flatten, SDense, GS2 | Flatten, Dense, BN, ReLU | 256 |
| | SDense, GS2 | Dense,BN, ReLU | 256 |
| | SDense | Dense | 22 |

The 1-Lipschitz version of resnet50 is described in Table 5. As the unconstrained version, It has around 25M parameters. For the large version, we simply multiply the number channels in hidden layers by $1.5$. The unconstrained version is the standart resnet50 architecture. In the case of imagenet we use the pretrained version provided by tensorflow.

Table 4: FashionMNIST Neural network architectures: Sconv2D is SpectralConv2D, GS2 is GroupSort2, SDense is SpectralDense, BN is BatchNorm, AvgPool is AveragePooling, SGAvgPool is ScaledGlobalAveragePooling (DEEL.LIP), GAvgPool is GlobalAveragePooling

| Dataset | OTNN | Unconstrained NN | |
| --- | --- | --- | --- |
| | Layer | Layer | Output size |
| FashionMNIST | Input | Input | $28 \times 28 \times 1$ |
| | SConv2D, GS2 | Conv2D, BN, ReLU | $28 \times 28 \times 96$ |
| | SConv2D, GS2 | Conv2D, BN, ReLU | $28 \times 28 \times 96$ |
| | SConv2D, GS2 | Conv2D, BN, ReLU | $28 \times 28 \times 96$ |
| | SConv2D (stride=2), GS2 | Conv2D (stride=2), BN, ReLU | $14 \times 14 \times 96$ |
| | SConv2D, GS2 | Conv2D, BN, ReLU | $14 \times 14 \times 192$ |
| | SConv2D, GS2 | Conv2D, BN, ReLU | $14 \times 14 \times 192$ |
| | SConv2D, GS2 | Conv2D, BN, ReLU | $14 \times 14 \times 192$ |
| | SConv2D (stride=2), GS2 | Conv2D (stride=2), BN, ReLU | $7 \times 7 \times 192$ |
| | SConv2D, GS2 | Conv2D, BN, ReLU | $7 \times 7 \times 384$ |
| | SConv2D, GS2 | Conv2D, BN, ReLU | $7 \times 7 \times 384$ |
| | SConv2D, GS2 | Conv2D, BN, ReLU | $7 \times 7 \times 384$ |
| | SGAvgPool | GAvgPool | 384 |
| | SDense | Dense | 10 |

Table 5: 1-lip resnet architecture for Imagenet and cat vs dog: Sconv2D is SpectralConv2D, GS2 is GroupSort2, SDense is SpectralDense, BC is Batchcentering (centeing without normalization), SL2npool is ScaledL2NormPooling2D, SGAvgl2Pool is ScaledGlobalL2NormPooling2D, GAvgPool is GlobalAveragePooling

| Layer | | | | output | |
|---|---|---|---|---|---|
| Input | | | | $224 \times 224 \times 3$ | |
| SConv2D 7-64 (stride=2), BC, GS2 | | | | $112 \times 112 \times 64$ | |
| InvertibleDownSampling | | | | $56 \times 56 \times 256$ | |
| $\begin{bmatrix} \text{SConv2D } 1 \times 1 & 64 & \text{BC, GS2} \\ \text{SConv2D } 3 \times 3 & 64 & \text{BC, GS2} \\ \text{SConv2D } 1 \times 1 & 256 & \text{BC} \\ \text{add-lip} & & \text{BC, GS2} \end{bmatrix} \times 3$ | | | | $56 \times 56 \times 256$ | |
| SL2npool | | | | $28 \times 28 \times 256$ | |
| $\begin{bmatrix} \text{SConv2D } 1 \times 1 & 128 & \text{BC, GS2} \\ \text{SConv2D } 3 \times 3 & 128 & \text{BC, GS2} \\ \text{SConv2D } 1 \times 1 & 512 & \text{BC} \\ \text{add-lip} & & \text{BC, GS2} \end{bmatrix} \times 4$ | | | | $28 \times 28 \times 512$ | |
| SL2npool | | | | $14 \times 14 \times 512$ | |
| $\begin{bmatrix} \text{SConv2D } 1 \times 1 & 256 & \text{BC, GS2} \\ \text{SConv2D } 3 \times 3 & 256 & \text{BC, GS2} \\ \text{SConv2D } 1 \times 1 & 1024 & \text{BC} \\ \text{add-lip} & & \text{BC, GS2} \end{bmatrix} \times 6$ | | | | $14 \times 14 \times 1024$ | |
| SL2npool | | | | $7 \times 7 \times 1024$ | |
| $\begin{bmatrix} \text{SConv2D } 1 \times 1 & 256 & \text{BC, GS2} \\ \text{SConv2D } 3 \times 3 & 256 & \text{BC, GS2} \\ \text{SConv2D } 1 \times 1 & 1024 & \text{BC} \\ \text{add-lip} & & \text{BC, GS2} \end{bmatrix} \times 3$ | | | | $7 \times 7 \times 2048$ | |
| SGAvgl2Pool | | | | 2048 | |
| SDense | | | | 1 
 1000 | cat vs dog 
 imagenet |

### A.2.3  Losses and optimizer

An extension of $\mathcal{L}^{hKR}$ to the multiclass case with $q$ classes. has also been proposed in [56] The idea is to learn $q$ 1-Lipschitz functions $f_1, \ldots, f_q$, each component $f_i$ being a *one-versus-all* binary classifier. The loss proposed was the following

$$\mathcal{L}^{hKR}_{\lambda,m}(f_{1,\ldots,q}) = \sum_{k=1}^{q} \left[ \mathop{\mathbb{E}}_{\mathbf{x} \sim \neg P_k} [f_k(\mathbf{x})] - \mathop{\mathbb{E}}_{\mathbf{x} \sim P_k} [f_k(\mathbf{x})] \right] + \lambda \mathop{\mathbb{E}}_{\mathbf{x},y \sim \bigcup_{k=1}^{q} P_k} (H(f_1(\boldsymbol{x}), \ldots, f_q(\boldsymbol{x}), y, m)$$

(4)

with :

$$H(f_1(\boldsymbol{x}), \ldots, f_q(\boldsymbol{x}), y, m) = (m - f_y(\boldsymbol{x}))_+ + \sum_{k \neq y} (m + f_k(\boldsymbol{x}))_+$$

This formulation has three main drawbacks: (i) for large number of classes several outputs may have few or no positive sample within a batch leading to slow convergence, (ii) weight of $f_y(\boldsymbol{x})$ (the function of the true class) with respect to the other decreases when the number of classes increases, (iii) the expectancy has to be evaluated through the batch, making the loss dependant of the size of the batch.

To overcome these drawbacks, we propose first to replace the Hinge term $H$ with a softmax weighted version. The softmax on all but true class is defined by:

$$\sigma(f_k(\boldsymbol{x}), y, \alpha) = \frac{e^{\alpha * f_k(\boldsymbol{x})}}{\sum_{j \neq y} e^{\alpha * f_j(\boldsymbol{x})}}$$

We can define a weighted version of $H$ function:

$$H_\sigma\left(f_1(\boldsymbol{x}),\ldots,f_q(\boldsymbol{x}),y,m,\alpha\right) = (m - f_y(\boldsymbol{x}))_+ + \sum_{k\neq y} \sigma(f_k(\boldsymbol{x}),y,\alpha) * (m + f_k(\boldsymbol{x}))_+$$

In this function, the value of $f_y(\boldsymbol{x})$ for the true class maintains consistent weight relative to the values of other functions, regardless of the number of classes. $\alpha$ is a temperature parameter. Initially, the softmax behaves like an average as all the values of $f_k$ are close. However, during the learning process, as the values of $|f_k|$ increase, the softmax transitions to function like a maximum. Similarly, if a low value is chosen for $\alpha$, the softmax behaves as an average, resulting in a one vs all hKR loss. By choosing a higher value for $\alpha$, the softmax unbalances the weights. Thus the loss persists as a one vs all hKR but incorporates a re-weighting of the opposing classes for each targeted class.

We also propose a sample-wise and weighted version of the KR part (left term in Eq 4). to get the proposed loss:

$$\mathcal{L}^{hKR}_{\lambda,m,\alpha}(f_{1,\ldots,q},x,y) = \left[\sum_{k\neq y}[f_k(\boldsymbol{x}) * \sigma(f_k(\boldsymbol{x}),y,\alpha)] - f_y(\mathbf{x})\right] \tag{5}$$
$$+ \lambda * H_\sigma\left(f_1(\boldsymbol{x}),\ldots,f_q(\boldsymbol{x}),y,m,\alpha\right)$$

It's important to note that this definition only applys to the balanced multiclass case (as in FashionMnist and ImageNet). In the unbalanced scenario, the weight must be rescaled according to the a priori distribution of the classes.

For CelebA, with hyperparameters $\lambda$ is set to 20, and $m = 1$. For FashionMNIST, we use Eq. 5, $\lambda$ is set to 5, $\alpha = 10$ and $m = 0.5$. For cat vs dog $\lambda$ is set to 10 and $m = 18$. For imagenet $\lambda$ is set to 500, $\alpha = 200$ and $m = 0.05$.

We train all networks with ADAM optimizer [39]. We use a batch size of 128, 200 epochs , and a fixed learning rate $1e{-}2$ for CelebA. For FashionMNIST we perform 200 epochs with a batch size of 128. We fix the learing rate to $5e{-}4$ for the 50 first epochs, $5e{-}5$ for the epochs 50-75, $1e{-}6$ for the last epochs. For cat vs dog we perform 200 epochs with a batch size of 256. We fix the learing rate to $1e{-}2$ for the 100 first epochs, $1e{-}3$ for the epochs 100-150, $1e{-}4$ for the epochs 150-180 and $1e{-}9$ for the last epochs. For imagenet we perform 40 epochs with a batch size of 512. We fix the learing rate to $5e{-}4$ for the 30 first epochs, $5e{-}5$ for the epochs 30-35, $1e{-}5$ for the epochs 35-38 and $1e{-}9$ for the last epochs.

## A.3 Complementary results

### A.3.1 FashionMNIST performances and ablation study

Table 6 presents different performance resuts on FashionMNIST. First line is the reference uncon-strained network. Second line shows the performances of the new version of $\mathcal{L}^{hKR}_{\lambda,\alpha}$. Table 6 also shows that the new version of the $\mathcal{L}^{hKR}_{\lambda,m,\alpha}$ in the multiclass case (Eq. 5) outperforms the $\mathcal{L}^{hKR}_{\lambda,m}$ defined in [56] (Eq. 4). Obviously, the accuracy enhancement is obtained at the expense of the robustness. The main interest of this new loss is to provide a wider range in the accuracy/robustness trade-off.

Table 6: FashionMNIST accuracy comparison with the different version of multiclass $\mathcal{L}^{hKR}_{\lambda,m}$. For the fixed margin, we use the one that performs best by parameter tuning (i.e. $m = 0.5$)

| Model | Accuracy |
|---|---|
| Unconstrained | 88.5 |
| OTNN $\mathcal{L}^{hKR}_{\lambda,m}$ multiclass version [56] ($\lambda = 10$, $m = 0.5$) | 72.2 |
| (Ours) OTNN $\mathcal{L}^{hKR}_{\lambda,m,\alpha}$ ($\lambda = 10$, $m = 0.5$, $\alpha = 10$) (Eq. 5) | **88.6** |

### A.3.2 CelebA performances

Table 7 presents the Sensibility and Specificity for each label reached by Unconstrained network and OTNN.

As a reminder, given True Positive (TP), True Negative (TN), False Positive (FP), False Negative (FN) samples, Sensitivity (true positive rate or Recall) is defined by:

$$Sens = \frac{TP}{TP + FN}$$

Specificity (true negative rate) is defined by:

$$Spec = \frac{TN}{TN + FP}$$

Table 7: CelebA performance results for unconstrained and OTNN networks

| Model | Metrics: Sensibility/Specificity | | | |
|---|---|---|---|---|
| | *Attractive* | *Bald* | *Big_Nose* | *Black_Hair* |
| Unconstrained | **0.83 / 0.81** | 0.64 / 1.00 | **0.65 / 0.87** | **0.74 / 0.95** |
| OTNN | 0.80 / 0.75 | **0.87 / 0.83** | 0.73 / 0.70 | 0.78 / 0.84 |
| | *Blond_Hair* | *Blurry* | *Brown_Hair* | *Eyeglasses* |
| Unconstrained | **0.86 / 0.97** | **0.49 / 0.99** | **0.80 / 0.88** | **0.96 / 1.00** |
| OTNN | 0.86 / 0.89 | 0.66 / 0.72 | 0.81 / 0.73 | 0.80 / 0.89 |
| | *Gray_Hair* | *Heavy_Makeup* | *Male* | *Mouth_Slightly_Open* |
| Unconstrained | 0.62 / 0.99 | **0.84 / 0.95** | **0.98 / 0.98** | **0.93 / 0.94** |
| OTNN | **0.84 / 0.83** | 0.89 / 0.83 | 0.92 / 0.89 | 0.80 / 0.89 |
| | *Mustache* | *Receding_Hairline* | *Rosy_Cheeks* | *Sideburns* |
| Unconstrained | 0.47 / 0.99 | 0.47 / 0.98 | 0.46 / 0.99 | **0.79 / 0.98** |
| OTNN | **0.86 / 0.76** | **0.81 / 0.79** | **0.82 / 0.80** | 0.79 / 0.82 |
| | *Smiling* | *Wearing_Earrings* | *Wearing_Hat* | *Wearing_Lipstick* |
| Unconstrained | **0.90 / 0.95** | **0.84 / 0.90** | **0.89 / 0.99** | **0.90 / 0.96** |
| OTNN | 0.84 / 0.88 | 0.78 / 0.72 | 0.86 / 0.90 | 0.90 / 0.89 |
| | *Wearing_Necktie* | *Young* | | |
| Unconstrained | 0.75 / 0.98 | **0.95 / 0.65** | | |
| OTNN | **0.87 / 0.86** | 0.79 / 0.69 | | |

### A.4 Complementary explanations metrics

#### A.4.1 Explanation attribution methods

An attribution method provides an importance score for each input variables $x_i$ in the output $f(x)$. The library used to generate the attribution maps is Xplique [24].

For a full description of attribution methods, we advise to read [21], Appendix B. We will only remind here the equations of

- Saliency: $g(\boldsymbol{x}) = |\nabla_{\boldsymbol{x}} f(\boldsymbol{x})|$
- SmoothGrad: $g(\boldsymbol{x}) = \underset{\delta \sim \mathcal{N}(0, \mathbf{I}\sigma)}{\mathbb{E}} (\nabla f(\boldsymbol{x} + \delta))$

SmoothGrad is evaluated on $N = 50$ samples on a normal distribution of standard deviation $\sigma = 0.2$ around $x$. Integrated Gradient [64], noted IG, is also evaluated on $N = 50$ samples at regular intervals. Grad-CAM [55], noted GC, is classically applied on the last convolutional layer.

#### A.4.2 XAI metrics

For the experiments we use four fidelity metrics, evaluated on 1000 samples of test datasets:

- Deletion [50]: it consists in measuring the drop of the score when the important variables are set to a baseline state. Formally, at step $k$, with $u$ the $k$ most important variables according to an attribution method, the Deletion$^{(k)}$ score is given by:

$$\text{Deletion}^{(k)} = f(\boldsymbol{x}_{[\boldsymbol{x}_u = \boldsymbol{x}_0]})$$

  The AUC of the Deletion scores is then measured to compare the attribution methods ($\downarrow$ is better). The baseline $x_0$ can either be a zero value (*Deletion-zero*), or a uniform random value (*Deletion-uniform*).

- Insertion [50]: this metric is the inverse of Deletion, starting with an image in a baseline state and then progressively adding the most important variables. Formally, at step $k$, with $u$ the most important variables according to an attribution method, the Insertion$^{(k)}$ score is given by:

$$\text{Insertion}^{(k)} = f(\boldsymbol{x}_{[\boldsymbol{x}_{\overline{u}} = \boldsymbol{x}_0]})$$

  The AUC is also measured to compare attribution methods ($\uparrow$ is better). The baseline is the same as for Deletion.

- $\mu$Fidelity [9]: this metric measures the correlation between the fall of the score when variables are put at a baseline state and the importance of these variables. Formally:

$$\mu\text{Fidelity} = \underset{\substack{u \subseteq \{1,\ldots,d\} \\ |u|=k}}{\text{Corr}} \left( \sum_{i \in u} g(\boldsymbol{x})_i, f(\boldsymbol{x}) - f(\boldsymbol{x}_{[\boldsymbol{x}_u = \boldsymbol{x}_0]}) \right)$$

  For all experiments, $k$ is equal to 20% of the total number of variables, and cutting the image in a grid of $20 \times 20$ ($9 \times 9$ for cat vs dog and imagenet). The baseline is the same as the one used by Deletion. Being a correlation score, we can either compare attribution methods, or different neural networks on the same attribution method ($\uparrow$ is better).

- Robustness-Sr [34]: this metric evaluate the average adversarial distance when the attack is done only on the most relevant features. Formally, given the $u$ most important variables:

$$\text{Robustness-Sr} = \left\{ \underset{\delta}{min} ||\delta|| \, s.t. argmax(f(\boldsymbol{x} + \delta)) \neq argmax(f(\boldsymbol{x})), \delta_{\overline{u}} = 0 \right\}$$

  where $\delta_{\overline{u}} = 0$ indicates that adversarial attack is authorized only on the set $u$. The AUC is measured to compare attribution methods ($\downarrow$ is better). Note this metric cannot be used to compare different networks, since it depends on the robustness of the network.

We use also several other metrics:

- Distances between explanations: to compare two explanation $f(x)$, we use either $L_2$ distance, or $1 - \rho$ where $\rho$ is the Spearman rank correlation [2, 23, 65] ($\downarrow$ is better).

- Explanation complexity: we use the JPEG compression size as a proxy of the Kolmogorov complexity ($\downarrow$ is better).

- Stability: As proposed in [74], the Stability is evaluated by the average distance of explanations provided for random samples drawn in a ball of radius $0.3$ ($0.15$ for cat vs dog and imagenet) around $x$. As before, the distance can be either $L_2$ or $1 - \rho$ ($\downarrow$ is better).

- Accuracy: To assess the relevance of explanation [57] use a semi-real dataset, called BlockMNIST, having a random *null block and evaluate the proportion of top-k feature attribution values within the null block.*

### A.4.3  Supplementary metric results

In this section we present several experiments and metrics that we were not able to insert in the core of the paper.

Deletion-zero and Insertion-zero are evaluated on CelebA and FashionMNIST dataset. It is known that the baseline value can be a bias for these metrics, and we are convinced that it has a higher influence with 1-Lipschitz networks. Even if results for Deletion-zero and Insertion-zero are less obvious than for Deletion and Insertion Uniform, we can see in Table 8, that for these metrics, the rank of Saliency is most of the time higher for OTNN.

Table 8: Insertion and Deletion metrics evaluation; GC: GradCam, GI: Gradient.Input, IG: Integrated Gradient, Saliency Rk : Rank (comparison by line only : in bold best score)

| Dataset | Network | Deletion-Zero ($\downarrow$ is better) | | | | | |
| | | GC | GI | IG | Rise | Saliency | SmoothGrad |
|---------|---------|-----|-----|-----|------|----------|------------|
| | | | | Deletion-Zero | | | |
| CelebA | OTNN | 8.01 | 7.04 | 7.05 | 7.09 | 6.98 (Rk2) | **6.96** |
| | Unconstrained | 5.77 | 4.56 | 4.38 | 5.07 | **4.13** (Rk1) | 4.51 |
| Fashion- | OTNN | 0.24 | 0.16 | **0.15** | 0.26 | 0.20 (Rk4) | 0.19 |
| MNIST | Unconstrained | 0.33 | 0.28 | 0.23 | **0.16** | 0.38 (Rk5) | 0.39 |
| | | | | Insertion-zero ($\uparrow$ is better) | | | |
| CelebA | OTNN | 10.26 | 11.63 | 11.58 | **15.50** | 10.06 (Rk6) | 10.10 |
| | Unconstrained | 14.24 | 11.71 | 12.37 | **15.70** | 6.67 (Rk6) | 7.65 |
| Fashion- | OTNN | 0.31 | 0.46 | **0.47** | 0.36 | 0.36 (Rk4) | 0.39 |
| MNIST | Unconstrained | 0.53 | 0.59 | 0.68 | **0.73** | 0.45 (Rk6) | 0.46 |

To leverage the bias of the baseline value, as proposed in [34] we evaluated the Robustness-SR metric, Saliency map on OTNN achieves top-ranking scores. One might argue that scores for unconstrained networks are lower, but this is directly linked to the higher intrinsic robustness of OTNNand thus cannot be compared.

Table 9: Robustness-SR metrics evaluation; GC: GradCam, GI: Gradient.Input, IG: Integrated Gradient, Saliency Rk : Rank (comparison by line only : in bold best score)

| Dataset | Network | Robustness-SR ($\downarrow$ is better) | | | | | |
| | | GC | GI | IG | Rise | Saliency | SmoothGrad |
|---------|---------|-----|-----|-----|------|----------|------------|
| CelebA | OTNN | 28.54 | 14.01 | 13.28 | 30.54 | **11.64** (Rk1) | 12.65 |
| | Unconstrained | 11.11 | 9.19 | 10.00 | 15.15 | 7.38 (Rk2) | **7.20** |
| Fashion- | OTNN | **1.69** | 3.31 | 3.36 | 3.27 | 2.29 (Rk3) | 2.01 |
| MNIST | Unconstrained | 1.17 | 1.36 | 1.17 | **1.15** | 1.21 (Rk4) | 1.25 |

The full results for the explanation complexity is given on Table 10. The complexity is still lower for OTNN on FashionMNIST, even if the gap with Unconstrained networks is narrower than for CelebA.

Accuracy was also assessed by learning an OTNN (MLP architecture) on BlockMNIST dataset [57], and evaluating the proportion of top-k saliency map values that fall in the *null* block. Fig 7 presents

Table 10: Complexity of Saliency map by JPEG compression (kB): lower is better

|               | CelebA | FashionMNIST |
|---------------|--------|--------------|
| OTNN          | 9.48   | 0.92         |
| Unconstrained | 16.84  | 0.94         |

several samples of input images and saliency maps of BlockMNIST dataset, to be compared to Fig.1 in [57]. It shows that OTNN gradients are almost only on the signal block (digit). And Table 11 evaluates the proxy metric proposed in [57], and confirms that OTNNs saliency maps top-k values point even more on discriminative features than those of adversarial trained networks.

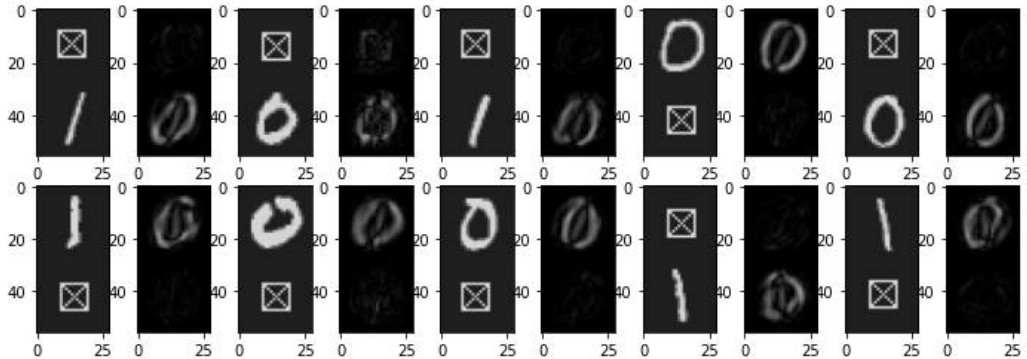

Figure 7: blockMNIST experiments using the proposed OTNN framework (10 blockMNIST images and their respective OTNN gradients). *null* marks are almost invisible in the gradients

Table 11: Comparison of Proxy metric on BlockMNIST (0 vs. 1) data for OTNNs, standard and robust models (values for the two last are directly extracted from [57])

| Unmasking fraction k | 2.5  | 5    | 10   | 15   | 20   | 25   | 30   |
|----------------------|------|------|------|------|------|------|------|
| Type of NN           | Fraction of top-k pixels in null block ||||||||
| Unconstrained [57]   | 43.8 | 42.5 | 44.8 | 46.5 | 47.5 | 48.1 | 48.4 |
| Adversarial [57]     | < 1  | < 1  | 2.3  | 7.9  | **16.7** | **24.2** | **29.9** |
| OTNN                 | < 1  | < 1  | **1.8** | **6.6** | 16.8 | 26.0 | 32.9 |

### A.5 Complementary qualitative results

In this section, we provide more samples of qualitative results and couterfactual exlanations for OTNN, based on the gradient, i.e. $x - t * \hat{f}(x)\nabla_x \hat{f}(x)$ for $t > 1$.

### A.5.1 FashionMNIST

Fig. 8 gives more results on FashionMNIST.

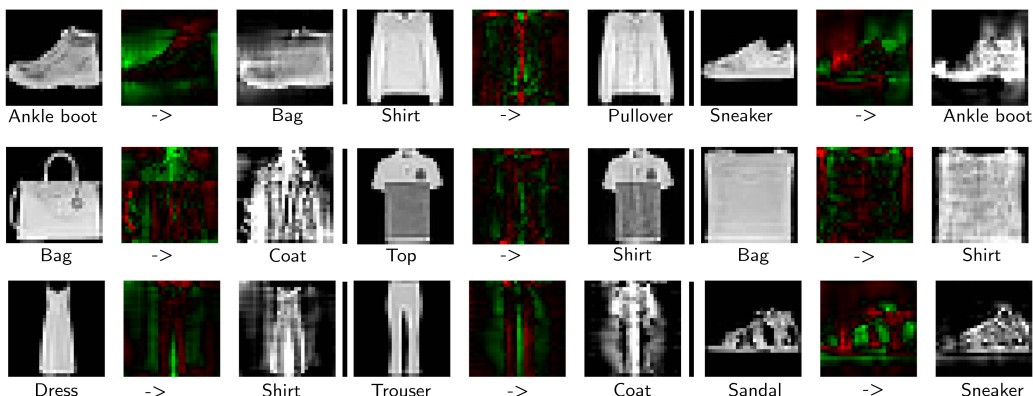

Figure 8: FashionMNIST samples

### A.5.2 CelebA

We presents results for other labels of CelebA. For ethic concerns we have hidden labels that can be subject to misinterpretation, such as *Attractive*, *Male*, *Big_Nose*. Fig. 9 to 27 present more results on the labels presented in the core of the paper, *Mouth_Slightly_Open*, *Mustache*, *Wearing_Hat*.

### A.5.3 Cat vs Dog

We present some supplementary comparison of Saliency Maps and counterfactual examples for cat vs dog(Fig. 28 and 29).

### A.5.4 Imagenet

Values of accuracy and human feature alignment used for the Fig.4 are described in Tab.12. We present some supplementary comparison of Saliency Maps Imagenet (Fig. 30). As pointed out previously, our model doesn't produce significant counterfacutal explanations on Imagenet.

Table 12: Comparison accuracy on human feature alignment of Saliency Maps different models on imagenet [22].

| Model | Accuracy | Human Aligment |
|---|---|---|
| clip | 56.0 | 0.03 |
| swin | 85.2 | 0.03 |
| vit_convnext | 85.8 | 0.15 |
| inception | 81.1 | 0.25 |
| resnet50_adv | 74.8 | 0.33 |
| resnet50 | 76.0 | 0.33 |
| VGG16 | 71.3 | 0.35 |
| resnet50_harmonized | 77.0 | 0.44 |
| OTNN | 67.0 | 0.54 |
| OTNN_large | 70.0 | 0.57 |

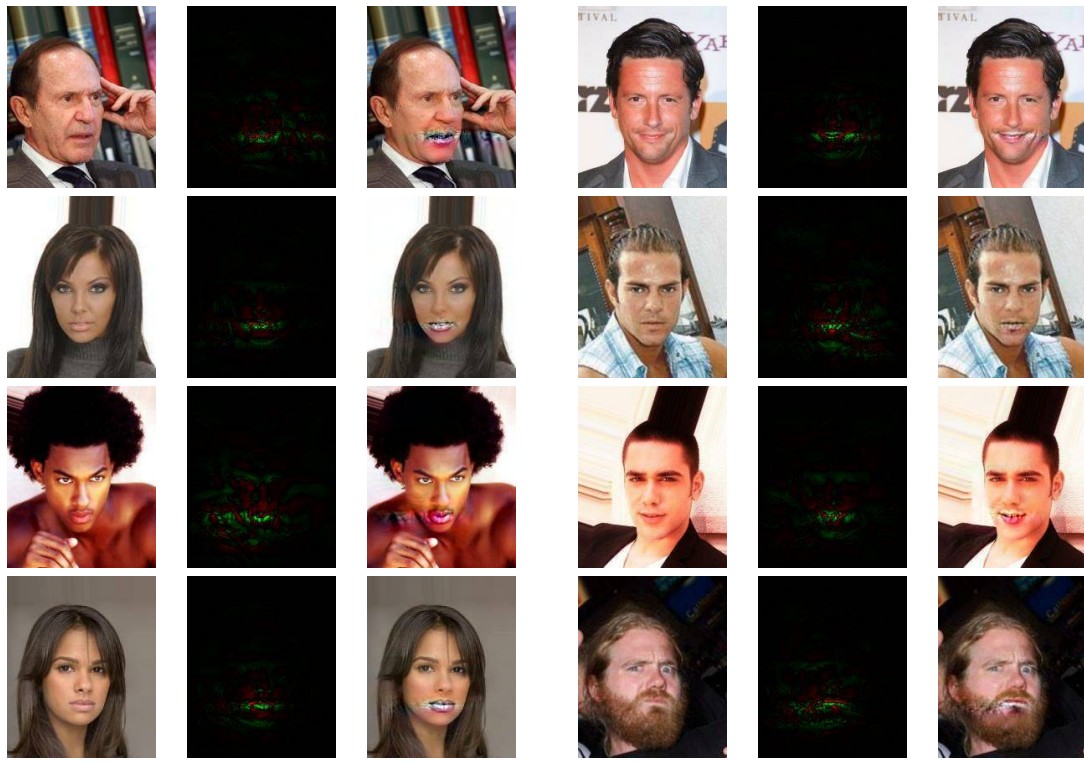

Figure 9: Samples from label Mouth_slightly_open: left source image (closed) , center difference image, right counterfactual (open) of form $\boldsymbol{x} - 10 * \hat{f}(\boldsymbol{x})\nabla_x \hat{f}(\boldsymbol{x})$

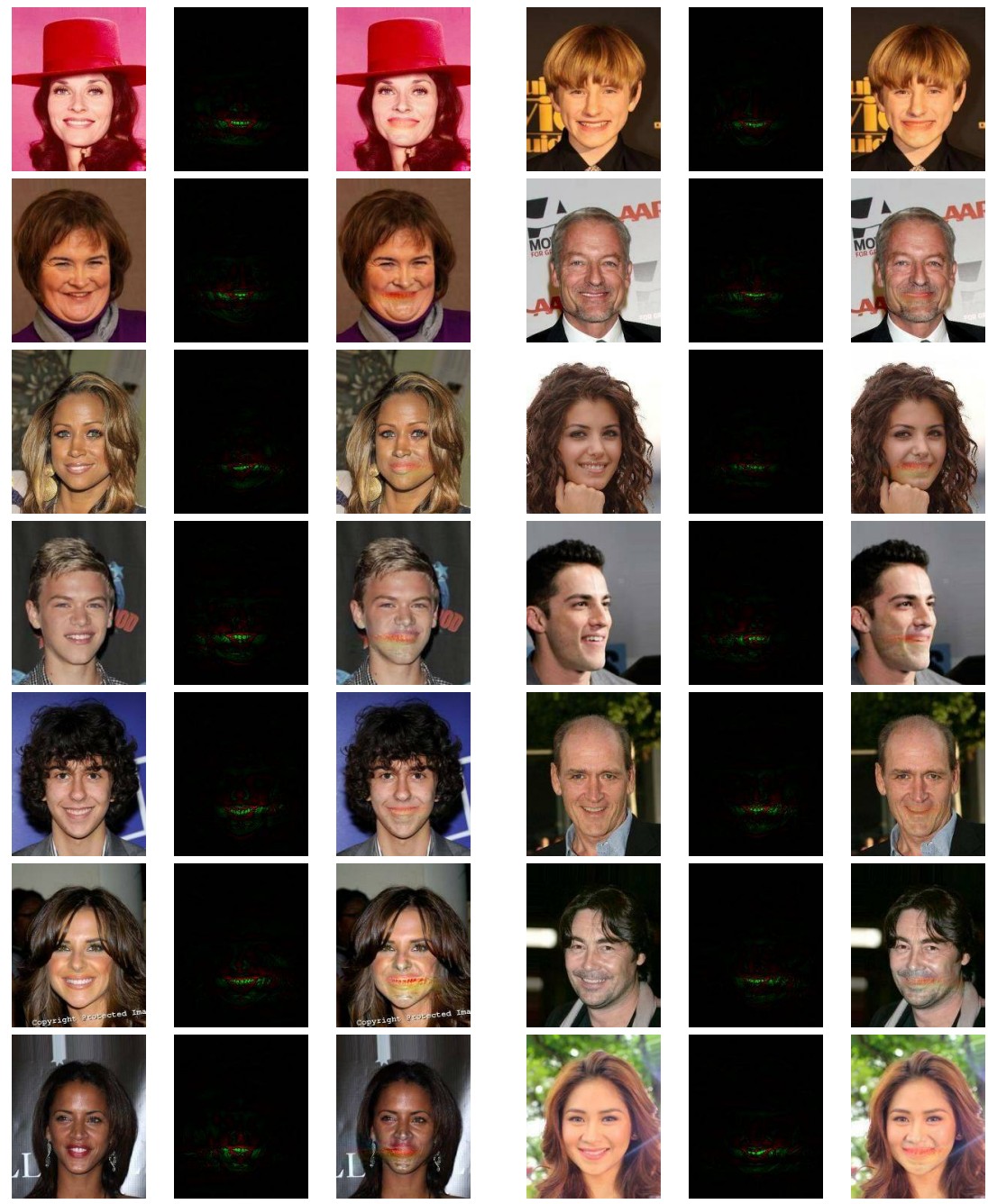

Figure 10: Samples from label Mouth_slightly_open: left source image (open) , center difference image, right counterfactual (close) of form $\boldsymbol{x} - 10 * \hat{f}(\boldsymbol{x})\nabla_x \hat{f}(\boldsymbol{x})$

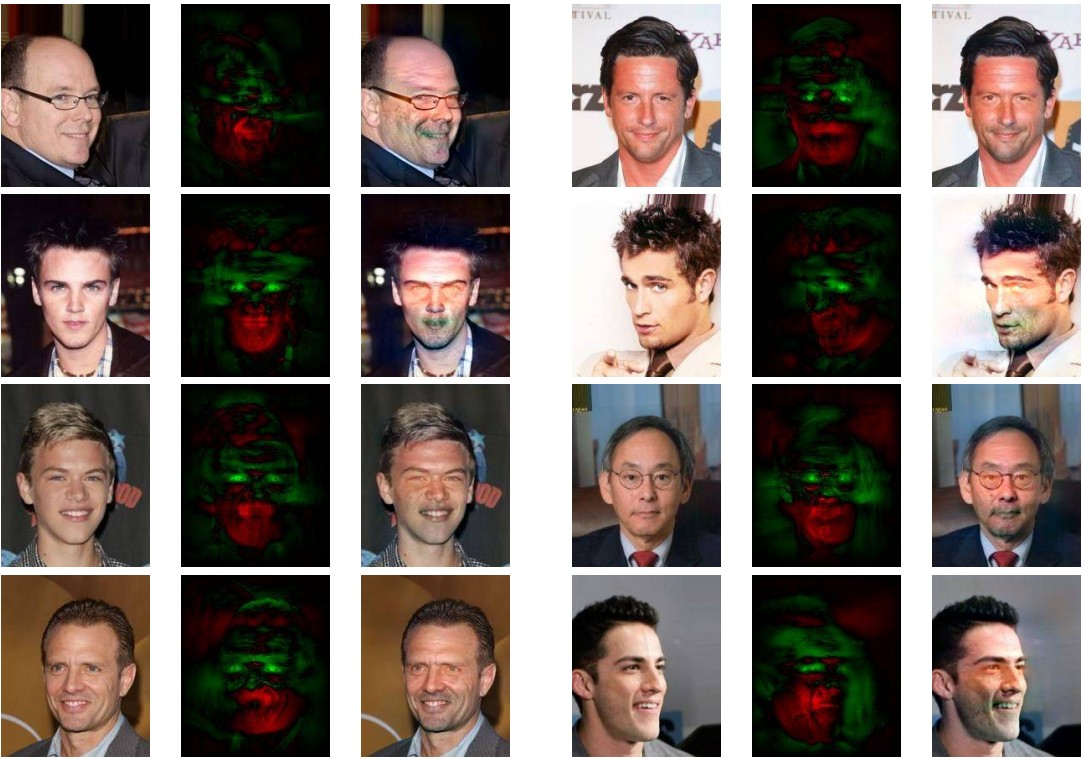

Figure 11: Samples from label Mustache: left source image (no mustache) , center difference image, right counterfactual (mustache) of form $\boldsymbol{x} - t * \hat{f}(\boldsymbol{x})\nabla_x \hat{f}(\boldsymbol{x})$ with $t \in \{5, 10, 20\}$

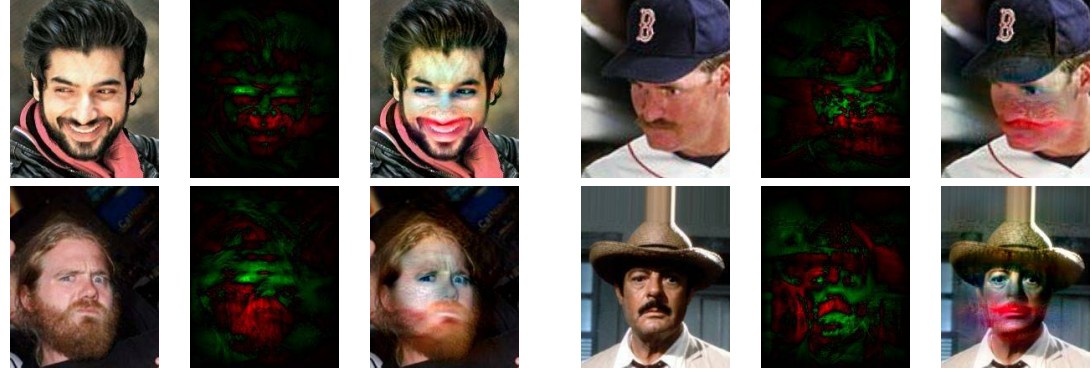

Figure 12: Samples from label Mustache: left source image (Mustache) , center difference image, right counterfactual (Non Mustache) of form $\boldsymbol{x} - t * \hat{f}(\boldsymbol{x})\nabla_x \hat{f}(\boldsymbol{x})$, $t \in 5, 10$

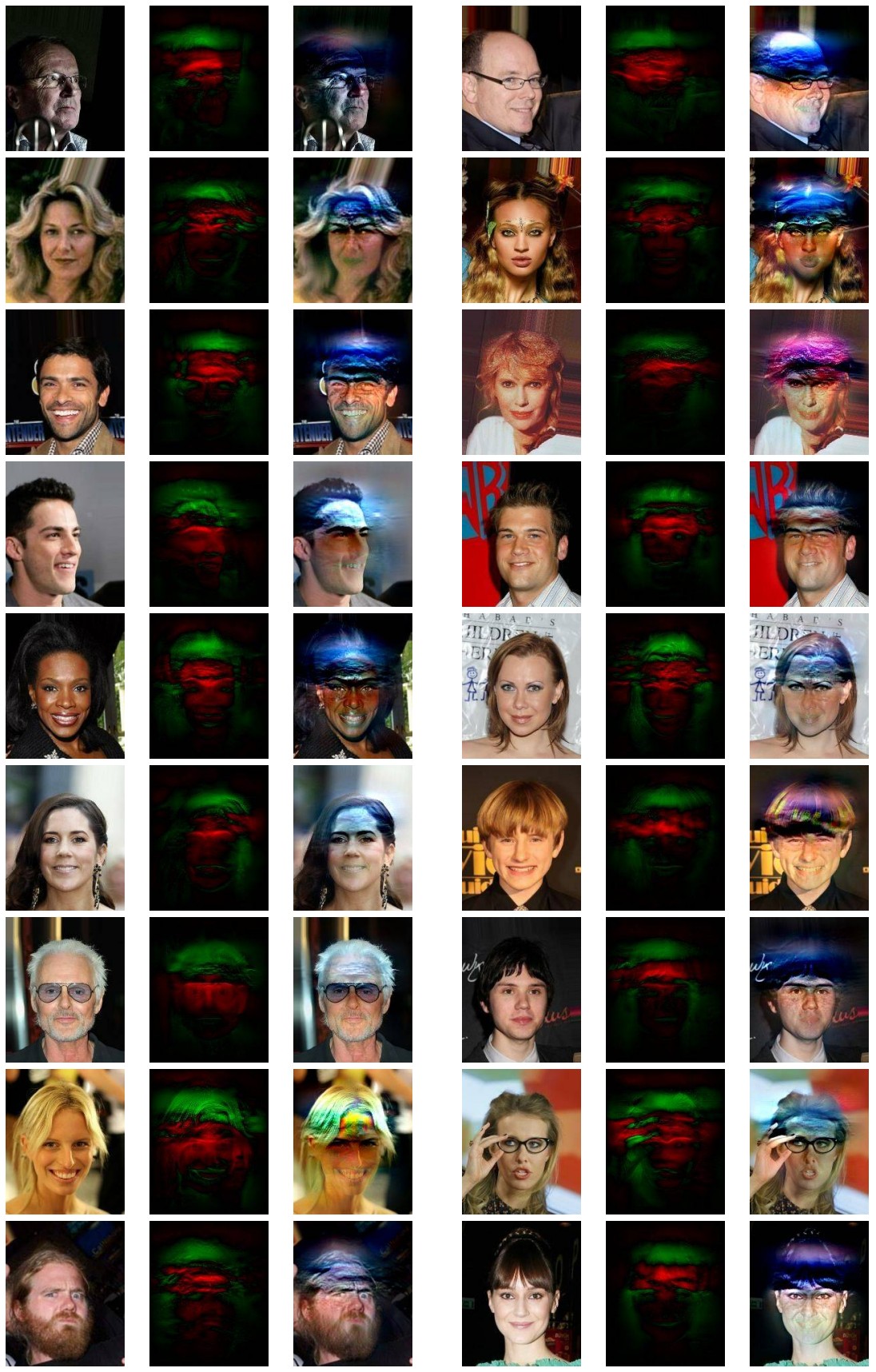

Figure 13: Samples from label Wearing Hat: left source image (No Hat) , center difference image, right counterfactual (Hat) of form $\boldsymbol{x} - t * \hat{f}(\boldsymbol{x})\nabla_x \hat{f}(\boldsymbol{x})$, $t \in 5, 10$

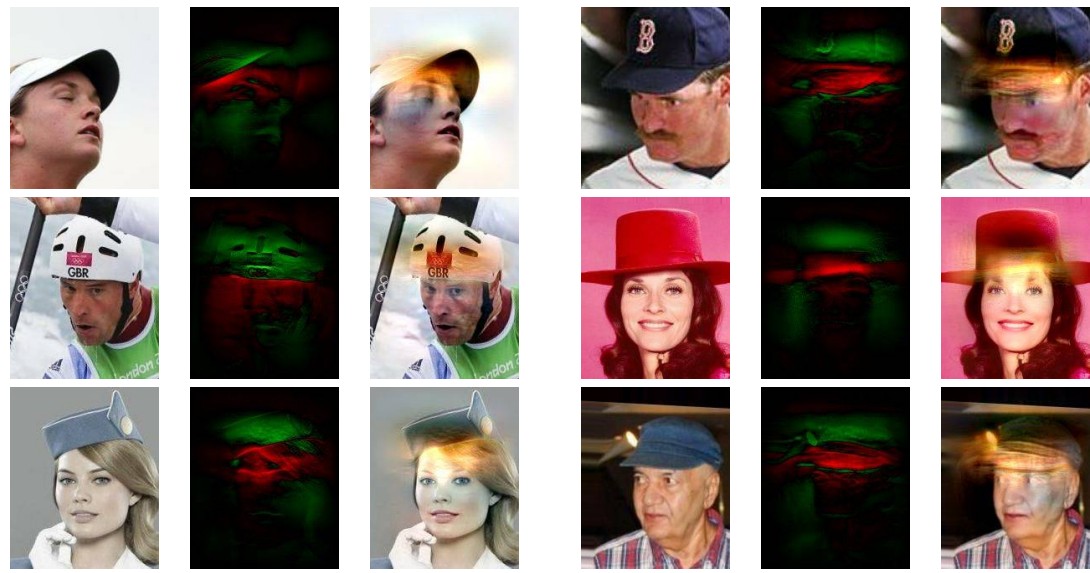

Figure 14: Samples from label Wearing Hat: left source image (Hat) , center difference image, right counterfactual (No Hat) of form $x - t * \hat{f}(x)\nabla_x \hat{f}(x)$, $t \in 5, 10$

Bald → "not" Bald

"not" Bald → Bald

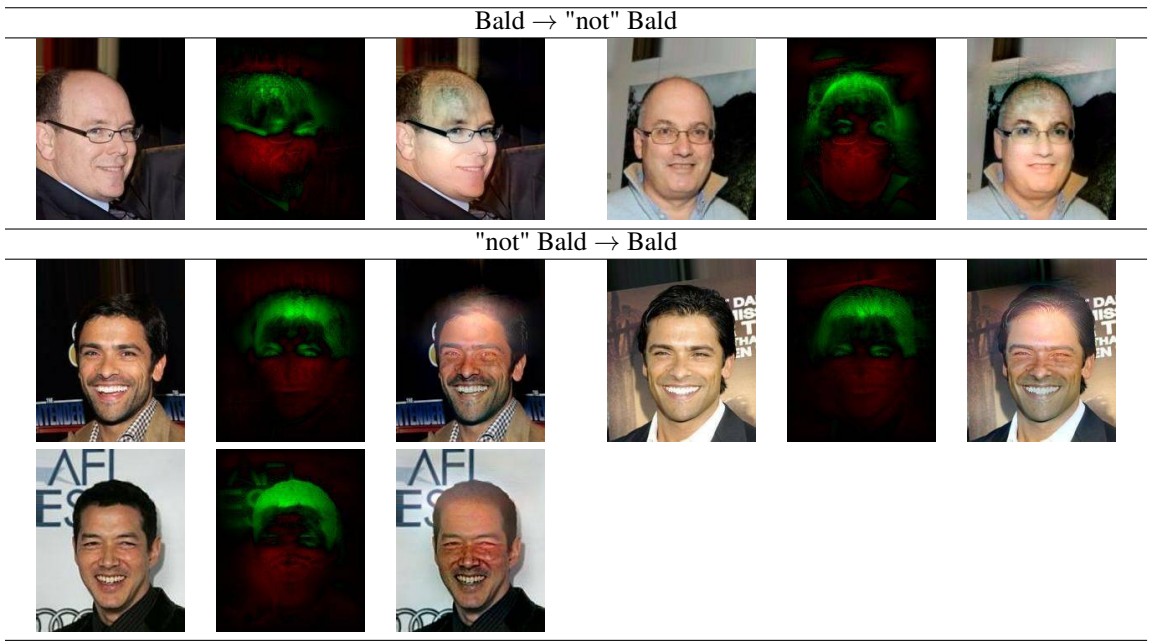

Figure 15: Samples from label Bald

Black_Hair → "not" Black_Hair

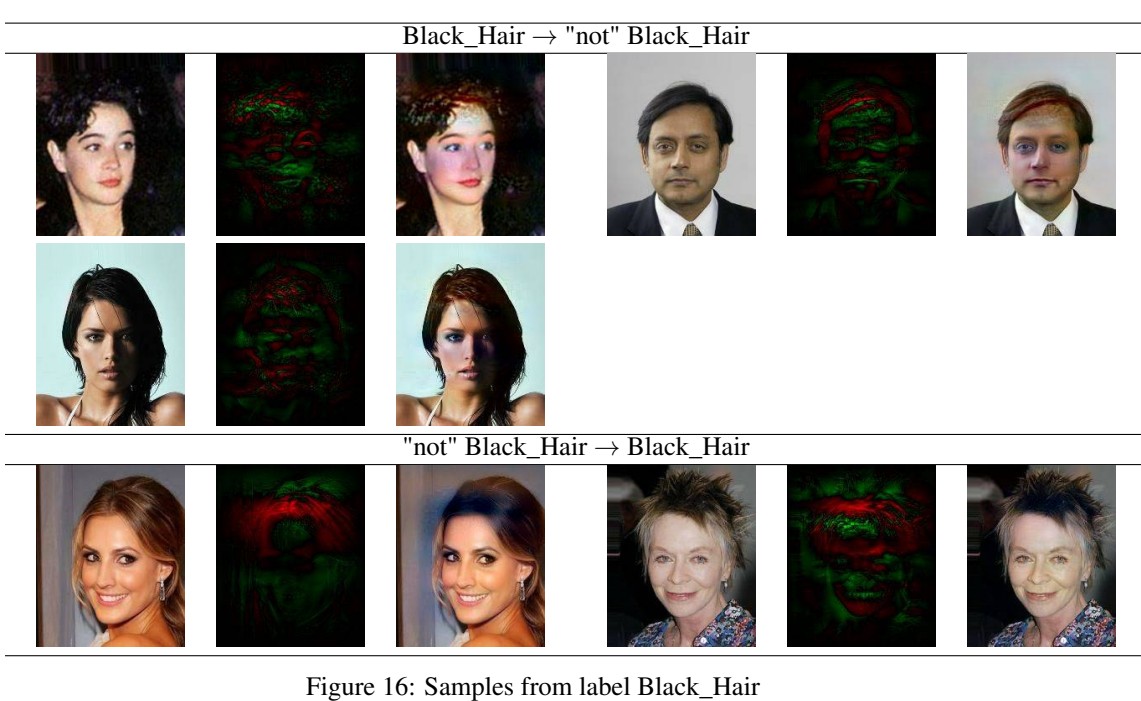

"not" Black_Hair → Black_Hair

Figure 16: Samples from label Black_Hair

Blond_Hair → "not" Blond_Hair

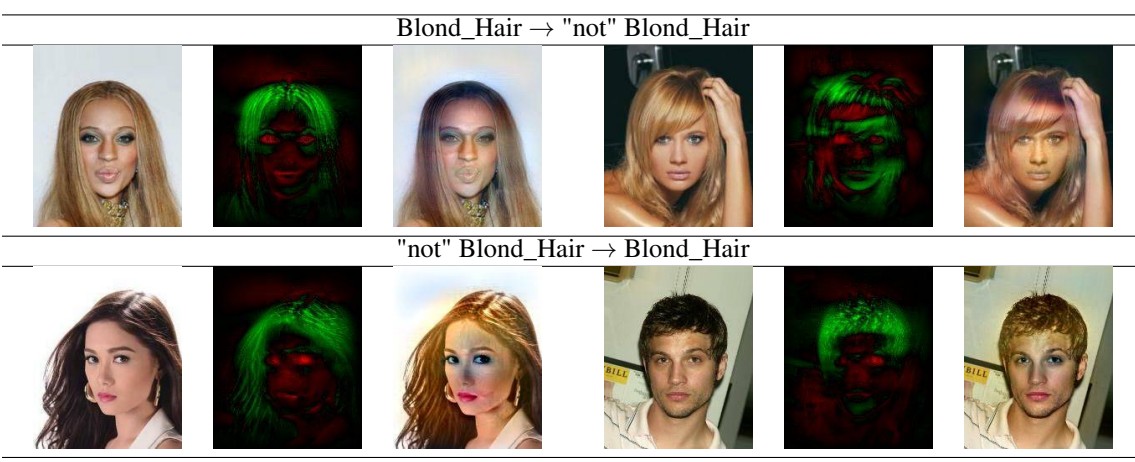

"not" Blond_Hair → Blond_Hair

Figure 17: Samples from label Blond_Hair

Blurry → "not" Blurry

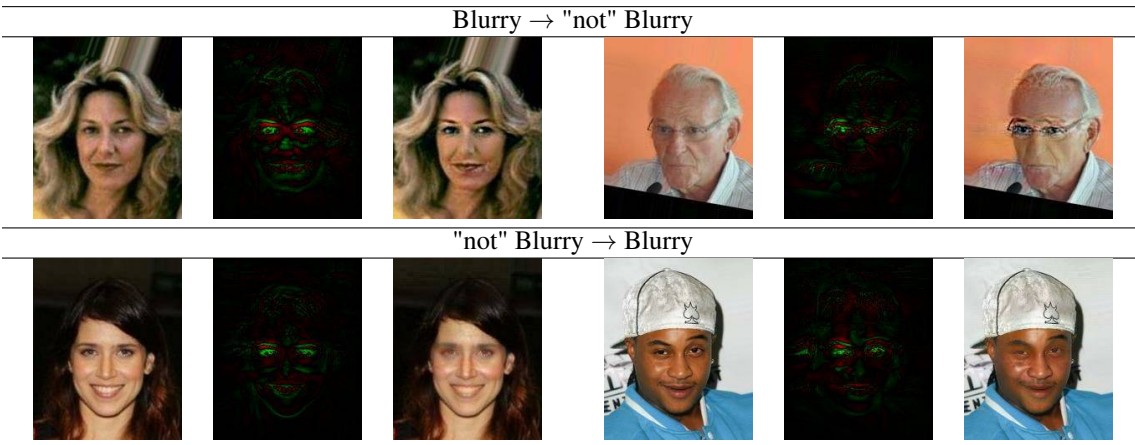

"not" Blurry → Blurry

Figure 18: Samples from label Blurry

Brown_Hair → "not" Brown_Hair

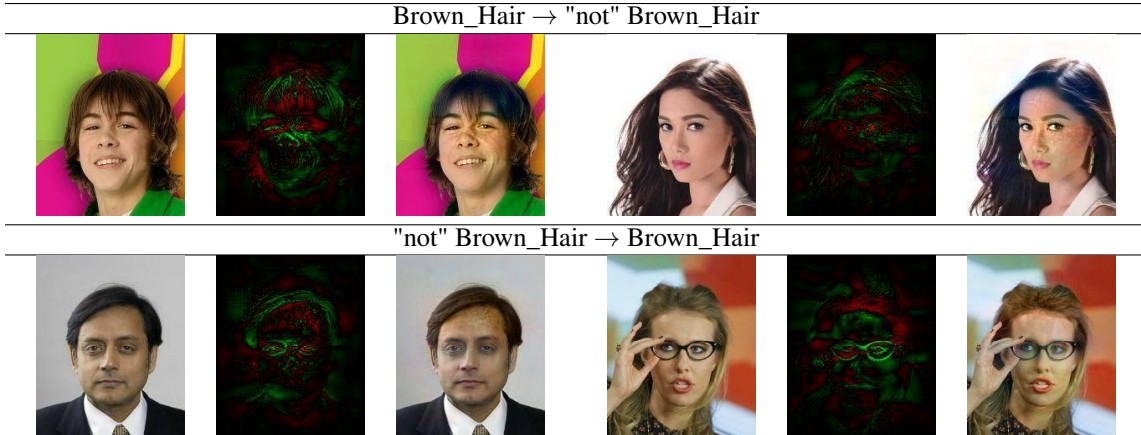

"not" Brown_Hair → Brown_Hair

Figure 19: Samples from label Brown_Hair

Eyeglasses → "not" Eyeglasses

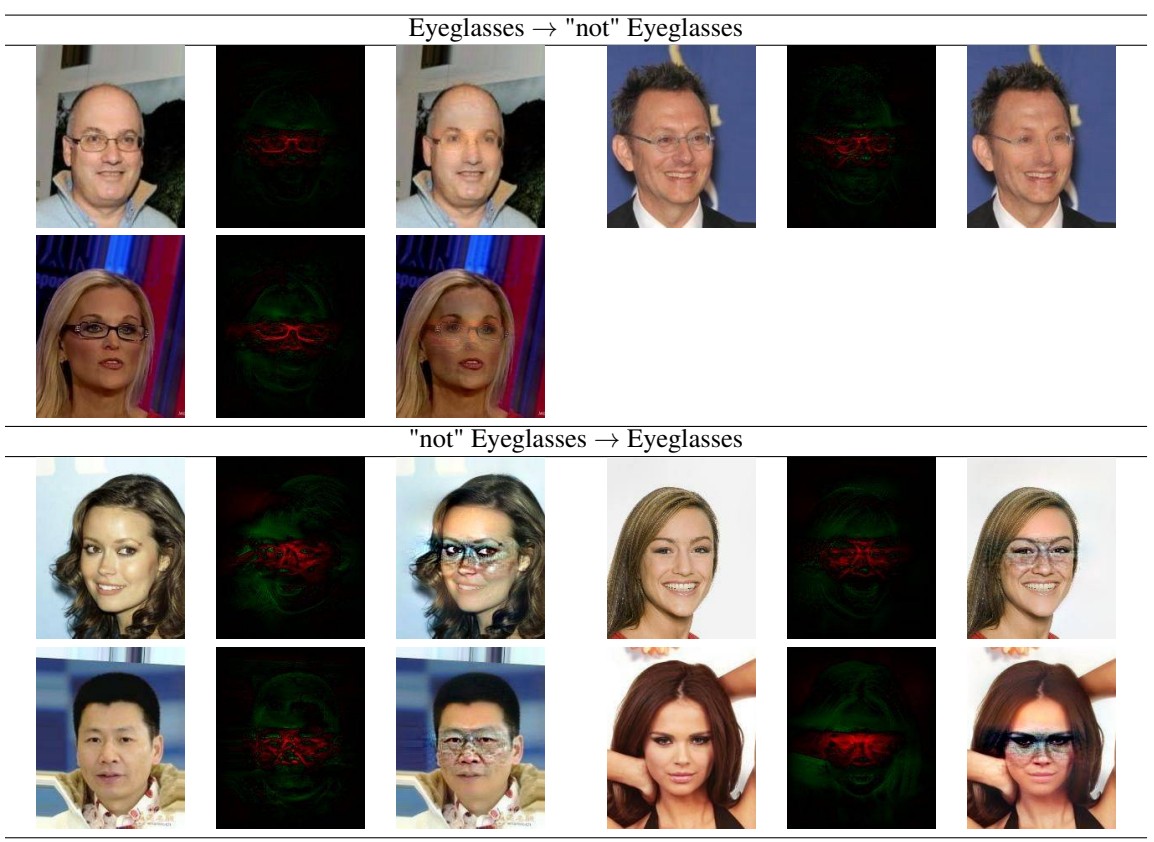

"not" Eyeglasses → Eyeglasses

Figure 20: Samples from label Eyeglasses

Gray_Hair → "not" Gray_Hair

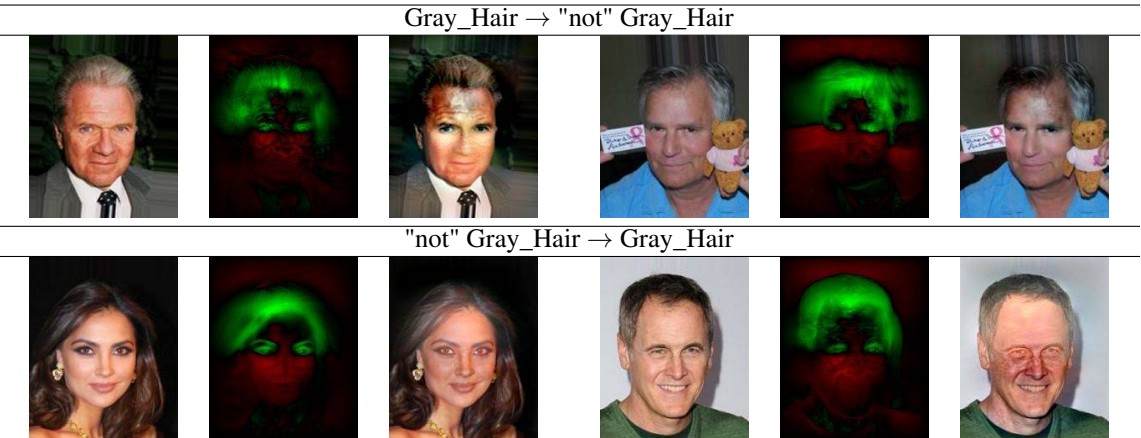

"not" Gray_Hair → Gray_Hair

Figure 21: Samples from label Gray_Hair

Hairline → "not" Hairline

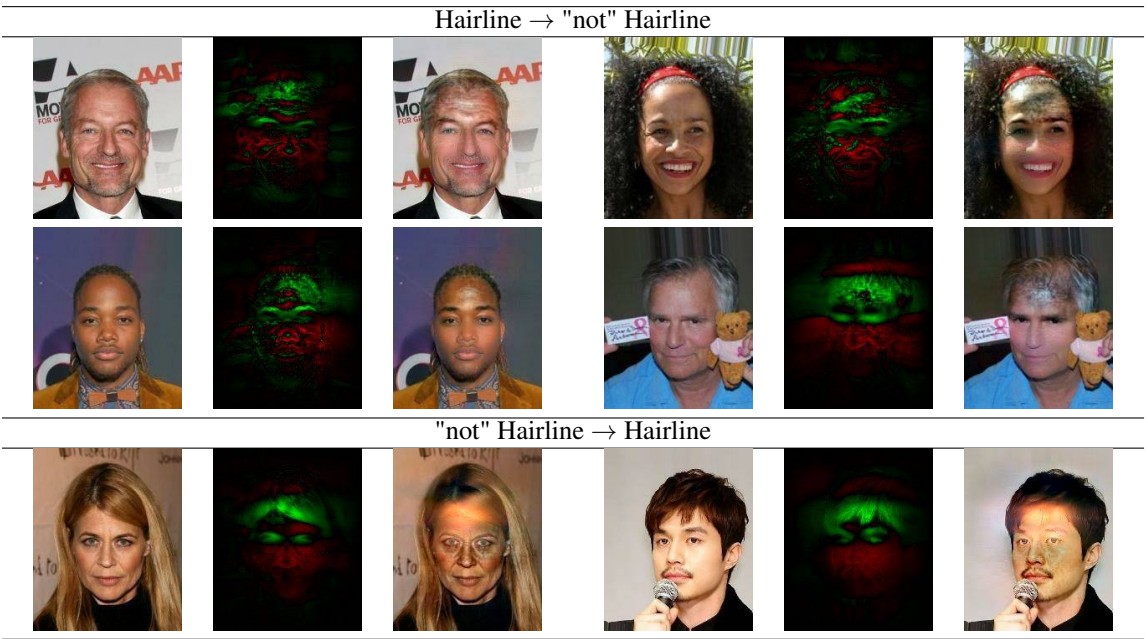

"not" Hairline → Hairline

Figure 22: Samples from label Hairline

Heavy_Makeup → "not" Heavy_Makeup

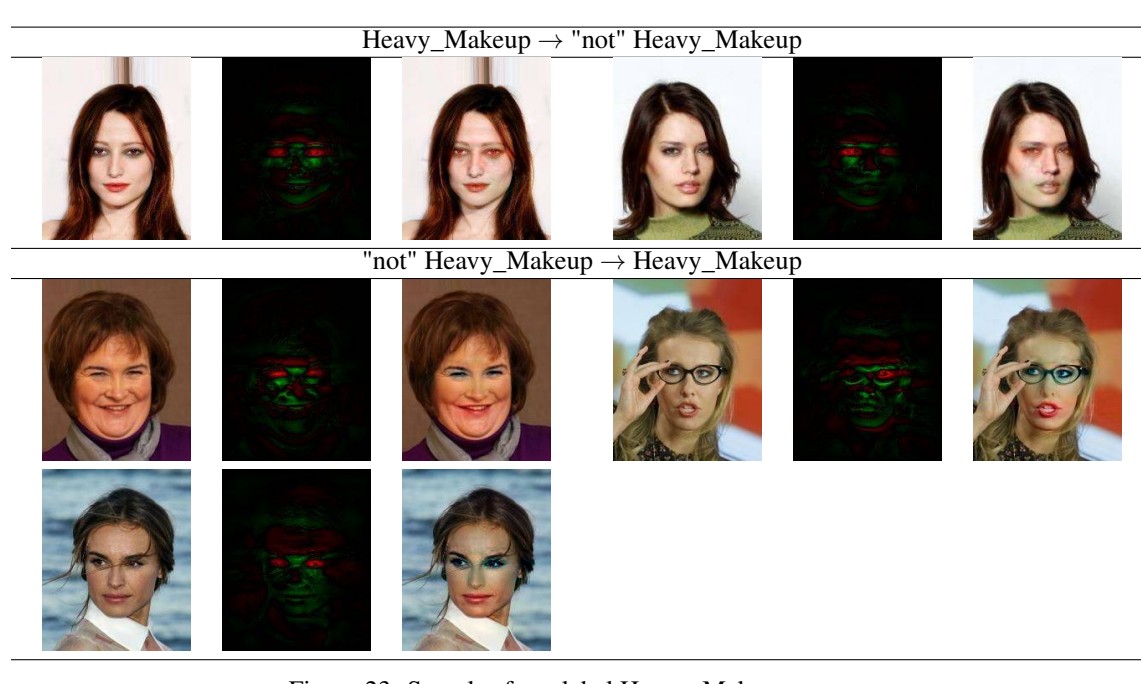

Figure 23: Samples from label Heavy_Makeup

Rosy_Cheeks → "not" Rosy_Cheeks

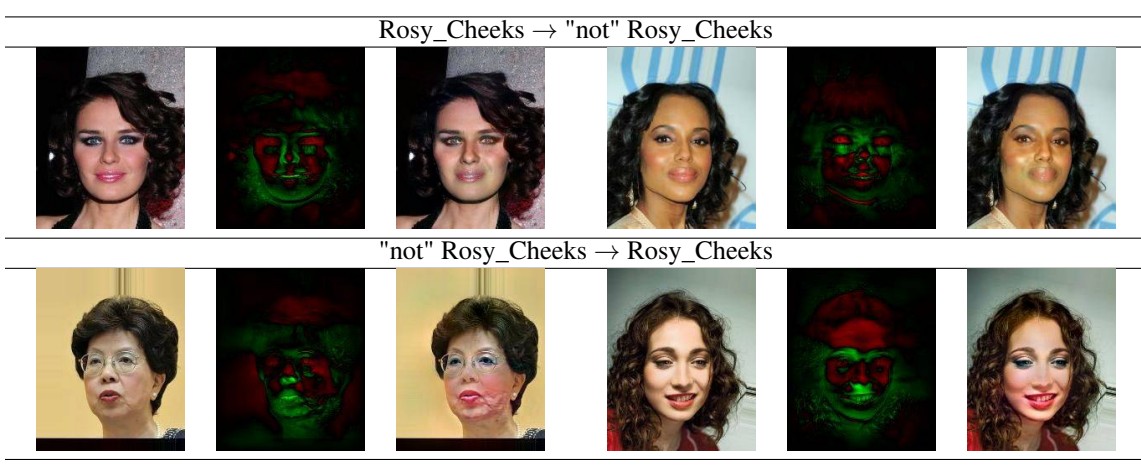

Figure 24: Samples from label Rosy_Cheeks

Smiling → "not" Smiling

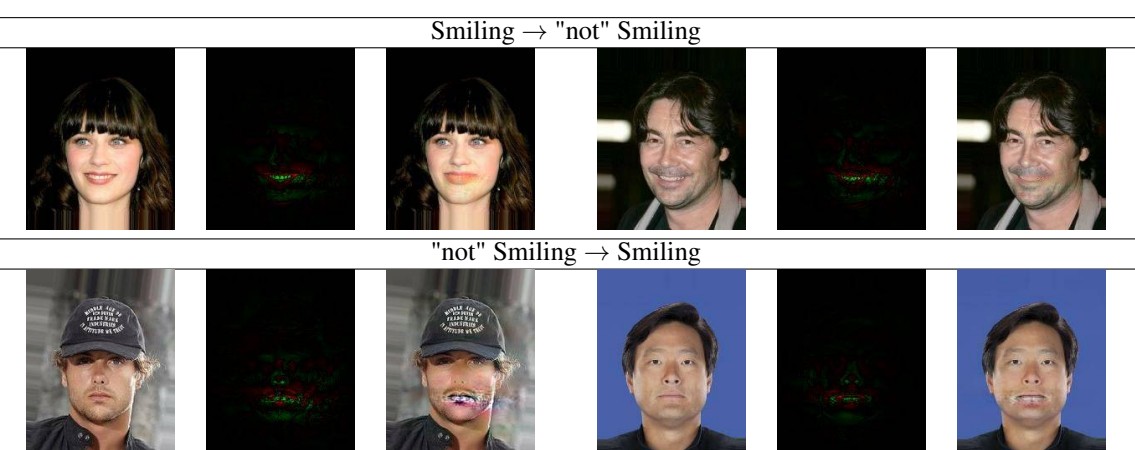

Figure 25: Samples from label Smiling

Wearing_Lipstick → "not" Wearing_Lipstick

"not" Wearing_Lipstick → Wearing_Lipstick

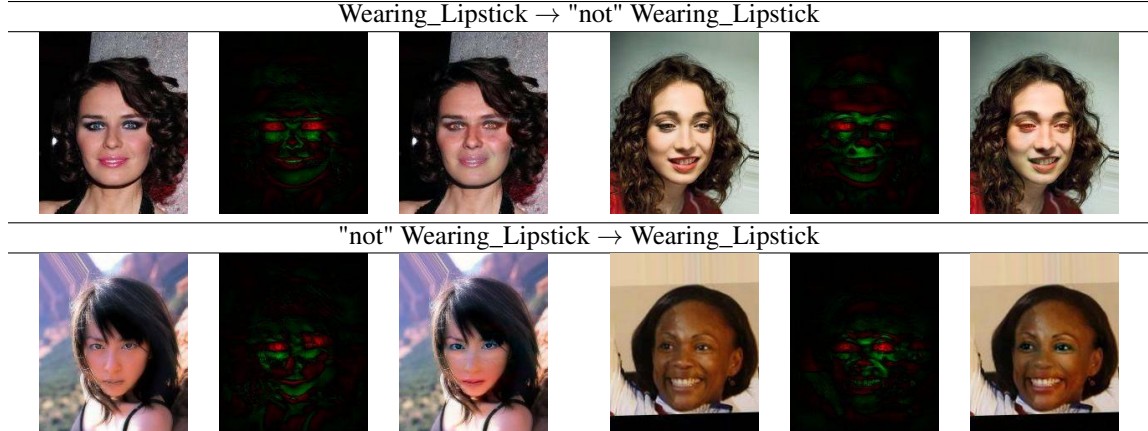

Figure 26: Samples from label Wearing_Lipstick

Young → "not" Young

"not" Young → Young

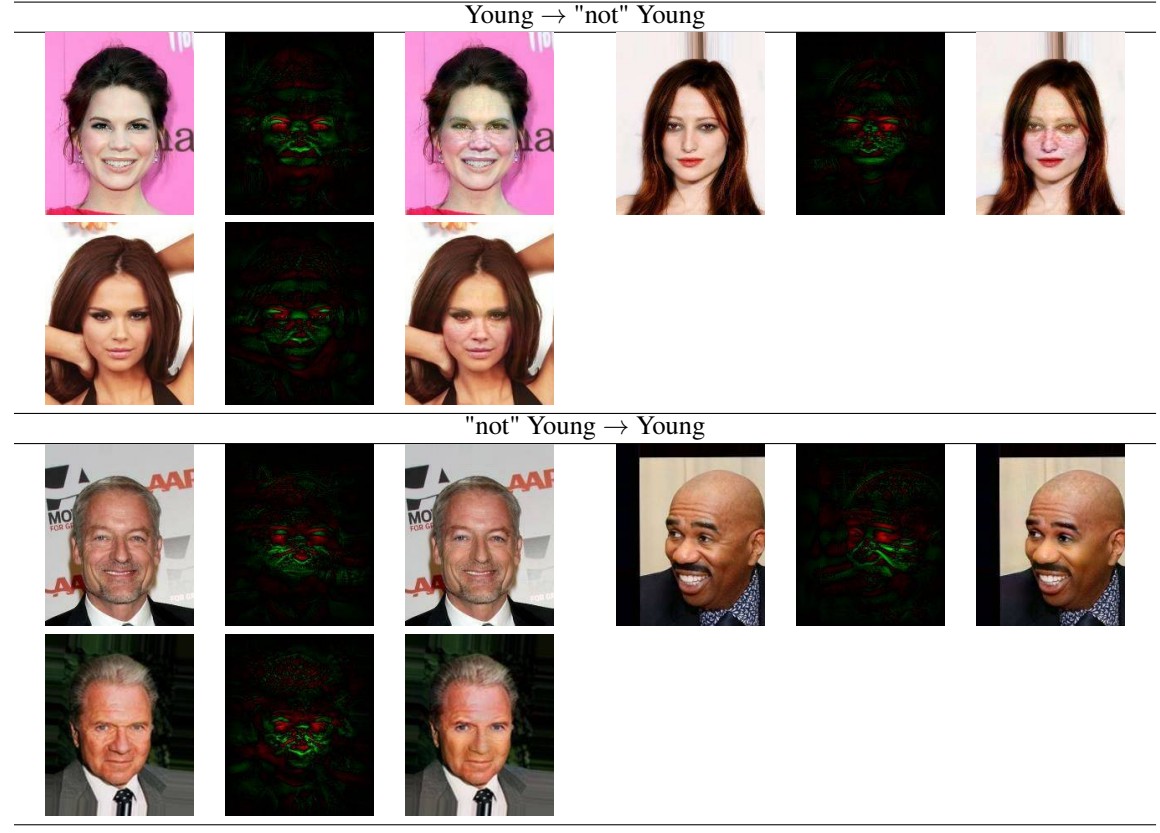

Figure 27: Samples from label Young

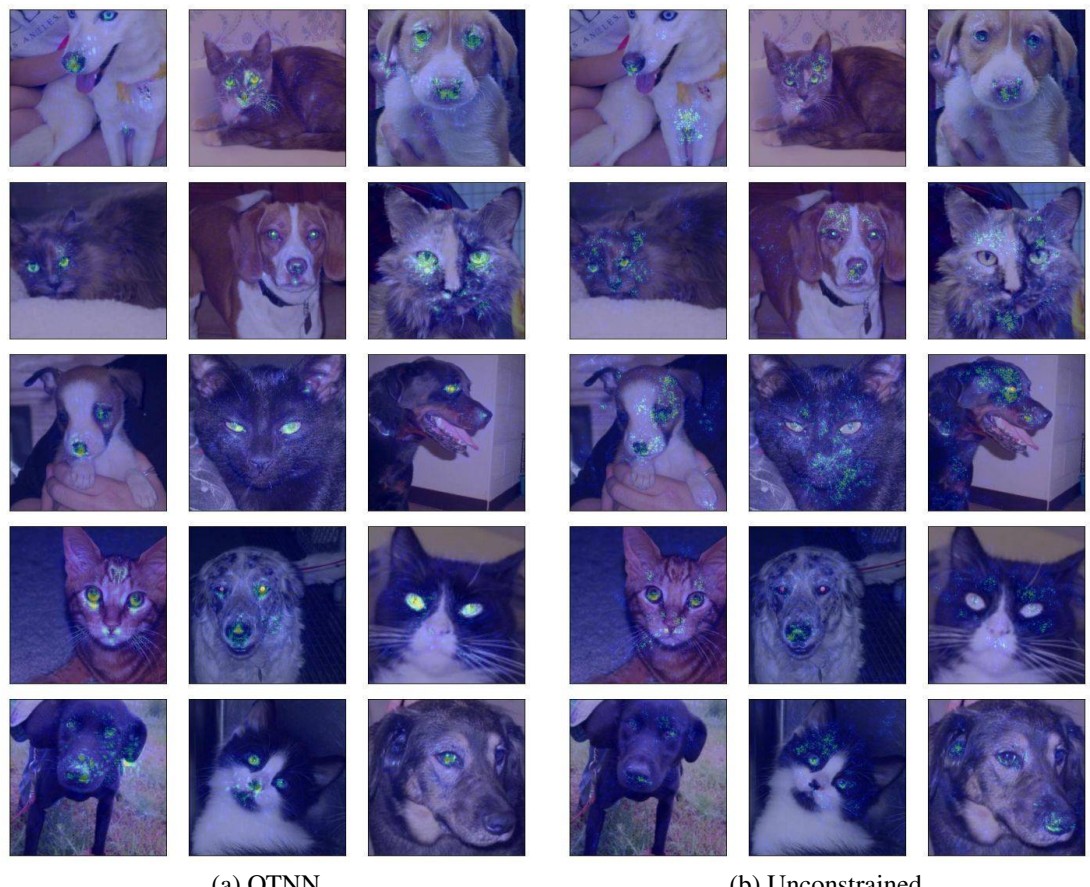

(a) OTNN                                    (b) Unconstrained

Figure 28: Cat vs Dog Saliency Map samples

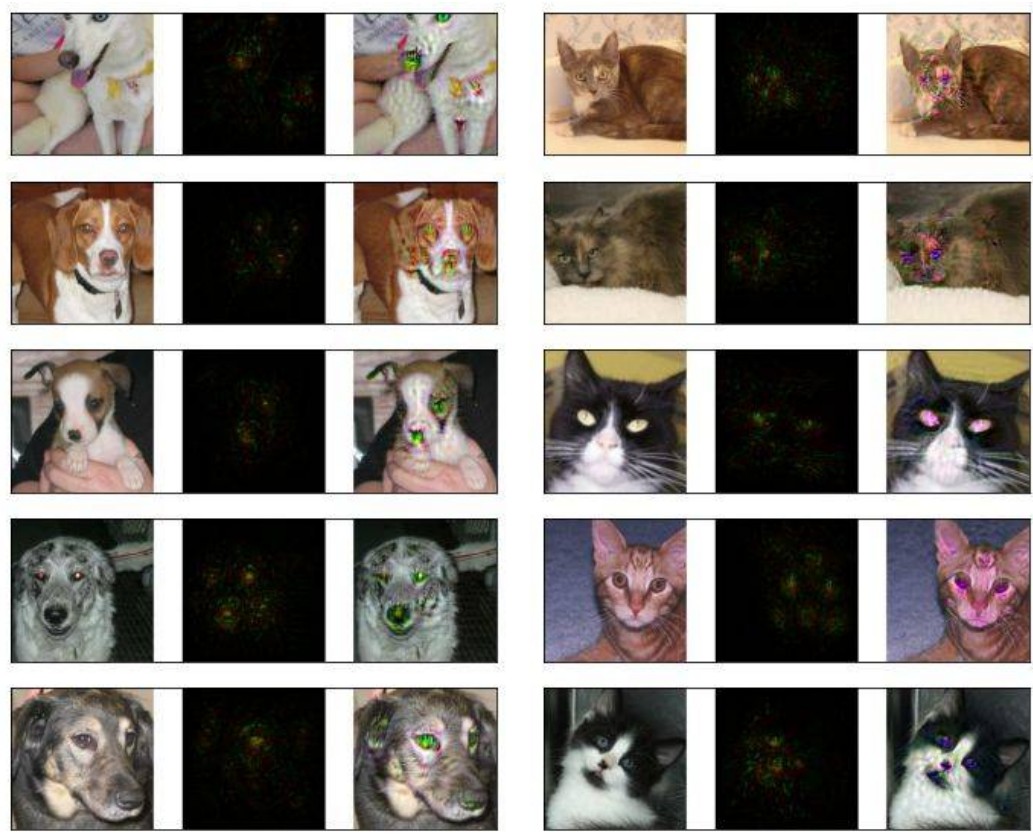

Figure 29: Cat vs Dog Saliency counterfactual samples. Left dog to cat, right cat to dog

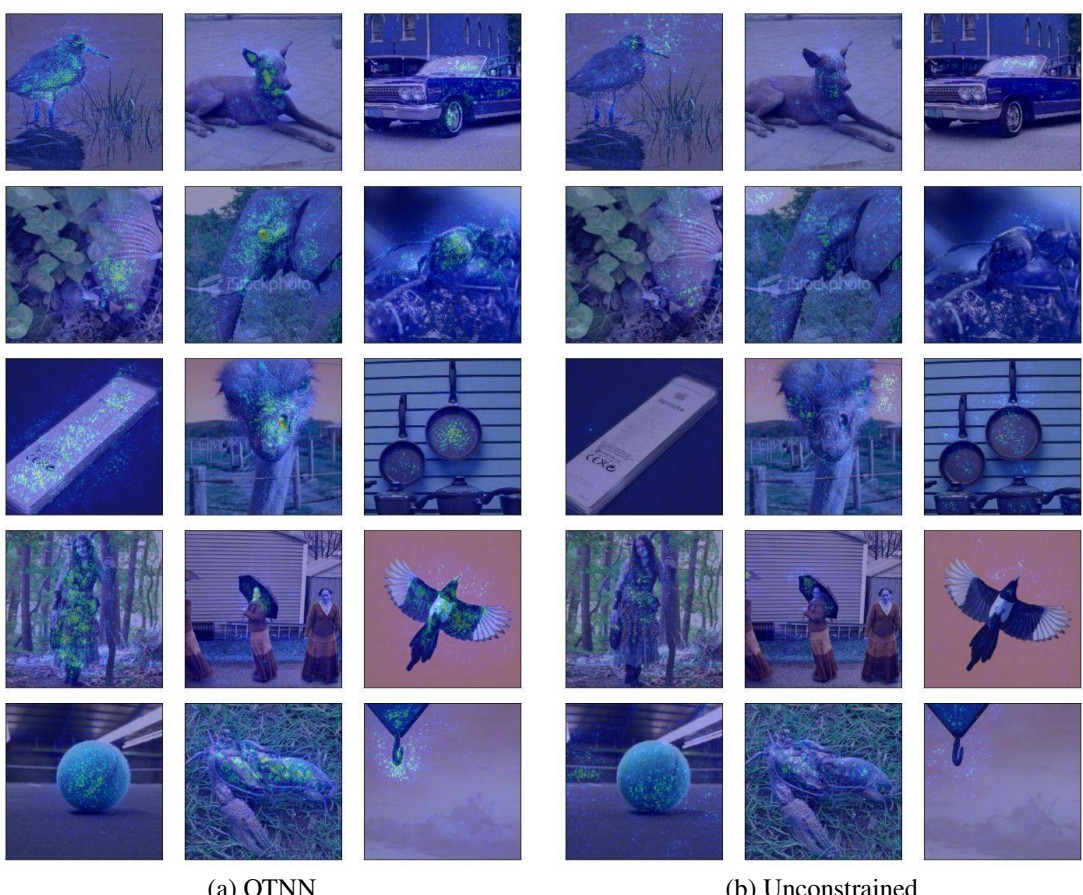

(a) OTNN              (b) Unconstrained

Figure 30: Imagenet Saliency Map samples

