# OpenReview forum: "On the explainable properties of 1-Lipschitz Neural Networks: An Optimal Transport Perspective"
_NeurIPS.cc/2023/Conference — NeurIPS 2023 poster_

### Official Review · Reviewer_n7GW · 2023-06-29

**Soundness:** 3 good
**Presentation:** 3 good
**Contribution:** 2 fair
**Rating:** 6
**Confidence:** 3

**Summary:**

In this paper, the authors provide a theoretical connection between 1-Lipschitz Net, also referred to as OTNN, with Optimal Transport, and provide various empirical results showing that gradient-based attributions look sharper and semantically meaningful.


**Strengths:**

One major contribution from the paper is Proposition 1 & 2, showing connections between the gradient direction and the optimal transport theory. The use of level curves aligns with my experience of analyzing properties of decision boundaries and smoothness of landscape. They use fidelity and stability metrics to show gradient-based attributions better capture the model’s behavior compared to standard (i.e. non-smooth) models. Also, I have not seen results using ClickMe and Feature Visualizations on OTTN, so they are also new empirical contributions to me.


**Weaknesses:**

The main weakness of this paper is its novelty or significance. This is because there are already a number of references, like [1, 2, 3], pointing out that robust models are more interpretable. Although many of these works target empirical robust models, 1-Lipschitz networks are simply the ones with stronger robustness guarantees: they are certifiably robust even if you do not enforce them (and will be more certifiably robust if you train them to do so) [4]. Also, there are works showing empirical robust training also improves the landscape smoothness by implicit gradient norm regularization [5]. Thus, connections between explainability and robustness based on 1-Lipschitz nets are only incremental and the insight is marginal from my personal perspective.

I appreciate two propositions that strengthen connections between robustness (more precisely the smoothness of the function) and explainability. I find the take-away regarding robustness and explainability is already well-known in 2023 (as the newest paper I refer to was published in 2021). This paper would be much more useful a few years ago to motivate the research on 1-Lipschitz networks; however, I think there are already a number of works pushing 1-Lipschitz nets forward. From my understanding, the current research problem is to close the gap between the theoretical possibility of learning a smoothing function and the practical difference of learning one on ImageNet-scale models and dataset. Because of the difficulty in scaling up 1-Lipschitz Net to larger datasets, it is actually more practical to do (empirical) adversarial training so we have a model that is both accurate and uses human-aligned features to make predictions.

[1] Christian Etmann, Sebastian Lunz, Peter Maass, and Carola Schoenlieb. On the connection between adversarial robustness and saliency map interpretability. In Proceedings of the 36th International Conference on Machine Learning, 2019.

[2] Francesco Croce, Maksym Andriushchenko, and Matthias Hein. Provable robustness of relu net- works via maximization of linear regions. AISTATS 2019, 2019.

[3] Wang, Z., Fredrikson, M., & Datta, A. (2021). Robust Models Are More Interpretable Because Attributions Look Normal. International Conference on Machine Learning.

[4] Trockman, A., & Kolter, J.Z. (2021). Orthogonalizing Convolutional Layers with the Cayley Transform. ArXiv, abs/2104.07167.

[5] Simon-Gabriel, C., Ollivier, Y., Bottou, L., Schölkopf, B. &amp; Lopez-Paz, D.. (2019). First-Order Adversarial Vulnerability of Neural Networks and Input Dimension. <i>Proceedings of the 36th International Conference on Machine Learning</i>, in <i>Proceedings of Machine Learning Research</i>


**Questions:**

My major question is pretty high-level: what are the insights from this paper that have not been covered by existing works.


**Limitations:**

Yes

---

> ### Author Rebuttal · Authors · 2023-08-03
>
>
> We greatly appreciate your thorough and insightful review, and have taken your valuable feedback into consideration. Please find below answers to your main questions:
> * Response to comment on “Related Work": We acknowledge in the paper that the link between robustness and explainability has been previously shown, citing three references [11,61,18].. We are open to incorporating additional papers suggested by the reviewer, given sufficient space in the final version. Similarly, we acknowledge that 1-Lipschitz Networks and Orthogonal layers are active research topics. However, our paper offers a substantial novelty with a theoretical proven link between orthogonality, 1-Lipschitz constraint, robustness, and explainability within the context of optimal transport. Adversarial training, while useful, lacks theoretical guarantees, and the improvements it offers in explainability do not match those provided by our approach. Furthermore, as illustrated in Figure 4 and [17], robust models (including some trained via adversarial training) do not necessarily enhance human alignment and can sometimes even degrade it.
> * Response to comment on “Novelty”: Neither the 1-Lipchitz constraint nor gradient $\nabla_x f=1$ almost everywhere independently suffice to ensure explainability. The Optimal transport loss, as described in line 311, “simultaneously addresses the classification task and induces a gradient alignment to the transportation plan,” directly impacting the explainability of the Saliency Map. This is a necessary condition for theoretical properties 1 and 2. This represents a novel and significant contribution from our perspective.
> * Response to comment on “difficulty in scaling up”:  We have trained the first certifiable 1-Lipschitz Neural Network on ImageNet with competitive accuracy (70%) and remarkable explainability properties, surpassing state-of-the-art models even at iso-accuracy (Figure 4). This is also an important contribution to the field.
> * Response to comment on “practical issues”: The OTNN approach is applicable to complex tasks (real-world applications are typically less complex than ImageNet) and comes with theoretical guarantees necessary in critical domains (such as transport and healthcare). We hope that it could convince those who care for guarantees that the impact of this approach is even higher than adversarial training methods.
>
> Collectively, these points underscore our viewpoint that the contribution is significant. We hope that it can convince you that this paper takes the field one step further to ‘close the gap’ and is more  than just ‘incremental or marginal’.

---

> > ### Comment · Reviewer_n7GW · 2023-08-11
> >
> > Thanks for the response for my question. My score remains positive (and stays at 6) after seeing the response.

---

> ### Author Response · Authors · 2023-08-11
> **Quick follow-up**
>
> Thank you for taking the time to review our work. We hope that our rebuttal has addressed your questions and concerns. Please let us know if you have any unresolved concerns or additional questions about the paper or our rebuttal

---

### Official Review · Reviewer_7q2B · 2023-07-04

**Soundness:** 2 fair
**Presentation:** 3 good
**Contribution:** 3 good
**Rating:** 5
**Confidence:** 4

**Summary:**

This paper focuses on a special type of 1-Lipschitz neural network named OTNN. The paper analyzes several theoretical properties of OTNN, including the correspondence between gradient direction and the optimal transportation plan from one class to another, the exact analytic solution of the decision boundary and the adversarial sample of OTNN. Furthermore, the paper shows that saliency maps of OTNN can provide meaningful explanation. Experiments also show that on various explanation methods, OTNN outperforms traditional neural networks in three commonly used XAI evaluation metrics.

**Strengths:**

1. The paper focuses on several interesting properties of OTNN, which have not been thoroughly investigated in previous works.

2. The paper is well-written and easy to follow.

**Weaknesses:**

1. About whether the OTNN model is locally linear. According to reference [48] in the main paper, the norm of the gradient $\nabla f^*(x)$ is 1 almost surely, and Corollary 1 implies that the adversarial sample is exactly located in the direction of the gradient. These properties seem to indicate that a trained OTNN model is nearly linear. Authors are suggested to clearly explain this issue. Note that I do not mean the whole decision boundary is linear, but a relatively large portion of local regions can be linear, as shown in Figure 2.

2. Current experimental results in the paper show that the performance of the OTNN model is far behind most existing models. Although I do admit that the performance of robust models like OTNN cannot compete with other neural networks, it is necessary to clearly and thoroughly compare the performance of OTNN and traditional neural networks in the main paper. Existing comparison experiments in the paper are not sufficient and are not well presented. Authors are encouraged to include large-scale comparison experiments between OTNN and traditional neural networks, organize the results into a table, and provide some discussion or comments for the comparison in the main text.

3. The dataset (the two concentric Koch snowflakes) used in the experiment in Figure 2 is too simple. The distribution of positive samples and negative samples in Figure 2 exhibit a high level of symmetry, which makes the decision boundary a regular hexagon. However, this toy dataset is not enough for the verification of theoretical results in Proposition 2 and Corollary 1. I suggest that authors conduct experiments on more complicated datasets to verify the theoretical results.

4. Authors are encouraged to visualize the loss landscape of the model trained on the two concentric Koch snowflakes in Figure 2, so as to verify the claim that gradient norm is 1 almost surely and the analytical solution of adversarial sample in Corollary 1 based on this visualization.



**Questions:**

1. It is encouraged to clarify and verify whether the trained OTNN model is almost linear in most of the local regions.
2. It is encouraged to have a thorough presentation and discussion in the main text about the comparison of classification performance between OTNNs and traditional neural networks.
3. It is encouraged to conduct experiments on more complicated datasets to verify the theoretical results in Proposition 2 and Corollary 1.

**Limitations:**

Yes.

---

> ### Author Rebuttal · Authors · 2023-08-03
>
> We greatly appreciate your thorough and insightful review, and have taken your valuable feedback into consideration. Please find below answers to your main questions:
> * Response to comment on “On almost linear property": Traditional deep learning models utilizing linear layers (Dense and Conv2D) and activations like ReLU are also piecewise linear functions. Imposing  $\nabla_x f=1$ almost everywhere only influences the slope of the prediction function in the direction of the decision boundary, a property that is certainly desirable. For instance, the Signed Distance Function outlined in [6] is $\nabla_x f=1$ almost everywhere and it is proven that it can approximate any decision boundary.
> * Response to comment on “On performance limitation": As stated in [6], 1-Lipschitz networks can be as accurate as unconstrained ones. Even though OTNN and the optimal transport loss impose guarantees that are stronger than a sole 1-Lipschitz constraint. Notably, robustness guarantees, and in our case the induced XAI guarantee, influence the accuracy. However, This impact remains negligible for all studied datasets (see lines 240-246), except for  ImageNet  on which the accuracy is clearly not in the tail of standard learnt models (see Fig 4) . A comprehensive comparison was performed to produce Figure 4, which includes over 100 state-of-the-art trained models (classical, adversarially trained, VGG, ResNet, and transformers). We propose to include the list of these networks, their accuracy, and alignment scores in the appendix and develop the discussion in the main text.
> * Response to comment on “Figure 2", the aim of this illustrative example was to sustain the two propositions rather than verify the model's performance. In Figure 2, all gradient norms are equal to 1 (we can include the numerical check in the final paper), and if they weren't, the shifted point $x-f(x)\nabla_x(f(x))$ wouldn't reside on the 0-level set. Thus, the figure perfectly illustrates propositions 1 and 2.
> * Response to comment on “More complex datasets", while we cannot sketch the boundary for such high dimensional datasets, we provide in Figure 5 and Appendix A.5 multiple samples for FashionMNIST, CelebA, CatvsDog, and ImageNet datasets, demonstrating that a single step in the gradient's direction  $x-f(x)\nabla_x(f(x))$ (equivalent to the black and red arrows in Fig2.b) is sufficient to perceptually alter the class (transport the input in direction to  another class). We believe these examples serve as generalizations of Figure 2 for more complex datasets.
> * Response to comment on “loss landscape ”: We dispute “weakness 3”, since the proposed properties are for the gradient of the input $x$ with respect to the logits $f(x)$, and not with respect to the loss (thus loss landscape is out of scope).
>
> We hope we have satisfactorily addressed your concerns. Happy to provide further clarifications if needed.

---

> > ### Comment · Reviewer_7q2B · 2023-08-17
> > **Responses to Authors**
> >
> > I have carefully read the rebuttal from the authors. My concerns regarding the performance and experimental verification in Figure 2 are addressed by the rebuttal. However, the concern regarding whether the trained OTNN is linear in a large local region remains unclear. Therefore, I would like to keep my original rating.

---

> > > ### Author Response · Authors · 2023-08-17
> > > **Re: Responses to Authors**
> > >
> > > Dear reviewer,
> > >
> > > We sincerely appreciate the time you've taken to review our rebuttal. We are pleased to note that we have addressed the majority of your concerns.
> > >
> > > We acknowledge that **your last concern (almost linear)  is more pertinent to the broader field and existing literature on OTNN, rather than the central theme of this paper**, which focuses on the explainability properties ascertained by the gradient direction and its alignment with human explanations.
> > >
> > > Anil et al.  [4] have demonstrated that such models (with $\nabla_x f=1$ almost surely) act as universal approximators for the space of 1-Lipschitz functions. Furthermore, Serrurier et al. [48] and Bethune et al. [6] have shown that such functions have no limitations when it comes to classification. Additionally, the OTNN encodes the potential function of a Wasserstein transportation plan are generally smooth functions (Villani [58]).
> > > On a more empirical note, we believe achieving a 70% accuracy on ImageNet would be unattainable with just few  linear separators.

---

> ### Author Response · Authors · 2023-08-11
> **Quick follow-up**
>
> Thank you for taking the time to review our work. We hope that our rebuttal has addressed your questions and concerns. Please let us know if you have any unresolved concerns or additional questions about the paper or our rebuttal

---

### Official Review · Reviewer_4Zh4 · 2023-07-05

**Soundness:** 3 good
**Presentation:** 3 good
**Contribution:** 3 good
**Rating:** 6
**Confidence:** 4

**Summary:**

This paper investigates the explainability properties of optimal transport neural networks, which are 1-Lipschitz constrained neural networks trained with an optimal transport dual loss. By establishing that the gradient of the optimal solution aligns with the transportation plan direction, and the closest decision boundary point (adversarial example) also lies in this gradient direction at a distance of the absolute value of the network output, the authors establish a link between the OTNN gradients and counterfactual explanations. They show that the gradient of OTNN significantly enhances the Saliency Map XAI method, improving several state-of-the-art XAI metrics.

**Strengths:**

- The paper is well motivated and well written.
- The problem tackled by the authors is interesting and important as the interpretability of neural networks is important for a large range of application
- The figures and empirical results are convincing

**Weaknesses:**

It appears that the authors use the DEEL.LIP library to construct the OTNN. The library was proposed by [1] and uses the power iteration to normalize each layer and constrain them to be 1-Lipschitz (see 4.1 of [1]). This approach to training 1-Lipschitz neural networks has recently been shown to be suboptimal [2, 3, 4], and new layers that are 1-Lipschitz _by design_ have been proposed. Can the authors comment on these papers, and can their results be improved with more recent 1-Lipschitz architectures?


[1] Serrurier et al. Achieving robustness in classification using optimal transport with hinge regularization, CVPR 2021
[2] Meunier et al., A Dynamical System Perspective for Lipschitz Neural Networks ICML 2022
[3] Araujo et al., A Unified Algebraic Perspective on Lipschitz Neural Networks, ICLR 2023
[4] Wang et al. Direct Parameterization of Lipschitz-Bounded Deep Networks, ICML 2023


**Questions:**

See weaknesses

**Limitations:**

1-Lipschitz networks are known to be highly constrained and do not provide state-of-the-art accuracy compared to their unconstrained counterparts. the authors should discuss this fact and the limitations of their work.

---

> ### Author Rebuttal · Authors · 2023-08-03
>
> We greatly appreciate your thorough and insightful review, and have taken your valuable feedback into consideration. Please find below answers to your main questions:
> * Response to comment on “1-Lipschitz layers": In fact, we implement a more complex approach than simple spectral normalization, as explained in appendix A.2.2 and using Deel.lip implementation. OTNNs necessitate $\nabla_x f=1$ almost everywhere, achievable via orthogonal kernels and GroupSort. Employing a more efficient algorithm to enforce the 1-Lipschitz and orthogonal constraints could be of interest, but it wouldn't change the theoretical results. It might, however, improve the experimental outcome scores without changing the conclusions. A comprehensive comparison of all these methods within the scope of explainability would warrant an independent paper altogether.
> * Response to comment on “Limitations of 1-Lipschitz NN": On the contrary, as proven in [6], 1-Lipschitz networks can achieve accuracy levels equivalent to unconstrained ones. However, OTNN and the optimal transport loss impose guarantees that exceed a simple 1-Lipschitz constraint. Particularly, it is recognized that robustness guarantees, and in our case the induced XAI guarantees, impact accuracy. This impact remains negligible for all studied datasets  (see lines 240-246), except for  ImageNet on which the accuracy is not in the tail of standard learnt models (see Fig 4) .

---

> > ### Comment · Reviewer_4Zh4 · 2023-08-10
> > **Follow-up questions**
> >
> > Thank you for your answers.
> >
> > I have a few follow-up questions.
> >
> > Indeed, your implementation does not use power iteration but orthogonal convolutions using the Bjorck algorithm [1], thank you for pointing that out. I would like to note that this approach has been used in [2] for orthogonalizing convolution in the context of adversarial robustness. This approach is known as BCOP. Another approach (with the Cayley transform) [3] has been proposed to orthogonalize convolutions and has been shown to outperform BCOP. Furthermore, more recent approaches (granted, they are not gradient preserving) have been developed to design 1-Lipschitz neural networks [4, 5, 6] and outperform both BCOP [2] and orthogonalization with the Cayley transform [3].
> >
> > I have the following questions:
> > - Could you explain why OTNN needs $\nabla_x f = 1$ and not just $\nabla_x f \leq 1$ ? (sorry if this question is already answered in the paper)
> > - I think your work could benefit from using more recent approaches for designing 1-Lipschitz networks, for example, if $\nabla_x f = 1$ is necessary, then using convolution with Cayley transform? or if $\nabla_x f \leq 1$ is sufficient, then using CPL, SLL or Sandwich?
> > - Regarding the implementation, I have read the code of the model "ResNet50_lip" in lip_res_model.py, could you explain how you maintain 1-Lipschitness with residual connections?
> >
> > There is a large body of work on 1-Lipschitz neural networks [2, 3, 4, 5, 6] that has not been discussed in the paper. I think it would be interesting to add a discussion of this related work.
> >
> >
> > [1] An Iterative Algorithm for Computing the Best Estimate of an Orthogonal Matrix
> > [2] Li et al., Preventing Gradient Attenuation in Lipschitz Constrained Convolutional Networks, NeurIPS 2019
> > [3] Trockman et al., Orthogonalizing Convolutional Layers with the Cayley Transform, ICLR 2021
> > [4] Meunier et al., A Dynamical System Perspective for Lipschitz Neural Networks ICML 2022
> > [5] Araujo et al., A Unified Algebraic Perspective on Lipschitz Neural Networks, ICLR 2023
> > [6] Wang et al. Direct Parameterization of Lipschitz-Bounded Deep Networks, ICML 2023

---

> > > ### Author Response · Authors · 2023-08-11
> > > **Re: Follow-up questions**
> > >
> > > Dear reviewer,
> > >
> > > We appreciate the extra time you've dedicated to reviewing our paper. Below, you will find responses addressing your additional questions:
> > >
> > > * Regarding the necessity of $\nabla_x f=1$: The work presented by Serrurier et al. [48] establishes that the HKR loss function is an optimal transport loss, with its solution being $\nabla_x f=1$ almost surely. The assertion of this fact is made in both the main body of the paper (line 121) and the appendix (as part of the proof of proposition 1). It is not a prerequisite for $\nabla_x f=1$ to be satisfied at each iteration of the optimization procedure; rather, the aim is to converge towards such a solution. In this context,  merely constraining the Lipschitz constant to 1 should be adequate. Nevertheless, our empirical observations demonstrate that striving to enforce $\nabla_x f=1$ for each layer as closely as feasible during the training process yields significantly improved results. A similar phenomenon has also been documented in Anil et al. [4]. While this particular insight is not expressly articulated in the body of the paper, we can add a  sentence for clarification.
> > > * Concerning the literature for 1-Lipschitz  for $\nabla_x f=1$: we agree and are aware of all the literature on lipschitz and orthogonal layers (some of them are cited in the paper). Employing a more efficient algorithm to enforce the 1-Lipschitz and orthogonal constraints could be of interest, but it wouldn't change the theoretical results of this paper. It might, however, improve the experimental outcome scores without changing the conclusions. Note that we have already assessed some alternatives to Bjork, such as Adam on the Riemannian manifold, AOL, and orthogonal regularization for convolution, without any significant improvement. This doesn't necessarily imply that Bjork is the best solution, but rather that tuning the interaction between the orthogonalization process and the optimization process is non-trivial. Performing an exhaustive comparative analysis of these methods within the scope of explainability would merit a distinct and standalone research publication. If required, we can add a sentence in the discussion part to mention this perspective.
> > > * Regarding the residual connection: Your observation is indeed accurate. The employed layer, represented as res = 0.5(x + g(x)) with $\nabla_x g=1$, satisfies the 1-Lipschitz criterion; however, there is no guarantee that it adheres to the condition $\nabla_x f=1$, for instance if  g = -Id. Nevertheless, since the optimal solution for the loss entails $\nabla_x f=1$, the learning process must inherently converge towards a solution aligned with this constraint. As a future work, it could be worthwhile to introduce a lipschitz-residual layer that enforces the constraint $\nabla_x res=1$, potentially leading to enhanced empirical performance metrics without altering the core conclusions.
> > >
> > > We hope that these explanations as the whole rebuttal will convince you that OTNN have interesting theoretical properties (robustness and explainability), and the contribution of this paper is significant. We believe it can stimulate a greater interest and open new perspectives to the field.

---

> > > > ### Comment · Reviewer_4Zh4 · 2023-08-14
> > > > **Follow-up questions**
> > > >
> > > > The authors addressed my concerns and provided further explanations when needed. I maintain my score.

---

### Official Review · Reviewer_WMVT · 2023-07-06

**Soundness:** 3 good
**Presentation:** 3 good
**Contribution:** 3 good
**Rating:** 6
**Confidence:** 4

**Summary:**

This paper studies the properties of 1-Lipschitz Neural Networks, and shows that the input-gradients of these models have desirable properties. On a benchmark across several datasets, they find that gradient-based feature attributions from the 1-Lipschitz Neural Networks show substantially better fidelity and stability metrics than those of an unconstrained model. Interestingly, for these networks, they show that adversarial attacks are counterfactual explanations (proposition 1). Overall, this paper adds to the line of work that shows that constrained (or regularized models) show improved input-gradients than unconstrained ones.

**Strengths:**

Overall, this is an interesting paper that explores the effect of constraining the Lipschitz constant of a model, and how it endows the model with reliable input-gradients.

**Quality/Clarity**\
Overall, this paper is quite well-written and clear. The key point of the paper is stated clearly and well-justified comprehensively across experiments on several datasets.

**Significance/Originality**\
I am combining the discussion on significance and originality because I think these are tightly linked for this work. First, I think several of the findings in this paper have been made in other papers (see weaknesses), but the specific settings under which this papers makes its findings are different. To the best of my knowledge, this is the first paper to explicitly and comprehensively quantify the fidelity and stability of explanations from constrained models. Overall, this paper is another important evidence to this line of work.


**Weaknesses:**

The main weaknesses are listed below:

**Lipschitz constant:** Why does it it have to be 1? Is there some range of values for this constant where the model gradients still retain these properties? How were these model trained? It says that they used some library, so I assume the training scheme here is standard and no new innovation there.

**Related work**: My main weakness of this paper has to do with related work. I'll list some here that the paper should engage with. First in [1], the authors show that the gradients of an adversarially robust model become sensitive to the output while those of an unconstrained model are not. In general, adversarially robust models have lower lipschitz constants, so the finding from this work makes sense. The authors should cite this paper. [2] and [3] also contain insights that add to this paper and explain some of their findings. For example that the model has better stability metrics is expected since that is essentially what constraining the lipschitz  constant does. I am not generally surprised by any of the insights discussed in this work, but this is fine, and they make a valuable contribution. However, it would be helpful if they could engage with the relevant literature below.

[1] Shah, Harshay, Prateek Jain, and Praneeth Netrapalli. "Do input gradients highlight discriminative features?." Advances in Neural Information Processing Systems 34 (2021): 2046-2059.

[2] Srinivas, Suraj, et al. "Efficient training of low-curvature neural networks." Advances in Neural Information Processing Systems 35 (2022): 25951-25964.

[3] Dombrowski, Ann-Kathrin, et al. "Towards robust explanations for deep neural networks." Pattern Recognition 121 (2022): 108194.

**Evaluation**: As the authors are probably aware, evaluating gradient attributions is tough. I looked into the fidelity and stability metrics, but I think these are merely passable. To really make concrete conclusions, the authors need a modified setting where ground-truth is known. I hate to be that reviewer that asks for new experiments, but something like the block MNIST evaluation in [1] is needed here to make conclusive claims. Lastly, the human evaluation using the click me dataset is nice, but addresses a different question. Just because an explanation aligns with what a human expects does not mean that the explanation is 'correct' or desirable. You want an explanation that tells you what your model has learnt. If your model has learnt human aligned signals then that is good. Comparing an explanations to humans is more helpful for debugging than for assessing the actual explanation itself.

**Questions:**

See weaknesses section for the questions that I have about this work.

Additional ones:

I assume Proposition 1 & 2 are new to this work? Or did they appear in previous work? I am not an expert on optimal transport, so I am less familiar with that literature.

**Limitations:**

The authors discuss societal impact in the conclusions section.

---

> ### Author Rebuttal · Authors · 2023-08-03
>
>
> We greatly appreciate your thorough and insightful review, and have taken your valuable feedback into consideration. Please find below answers to your main questions:
> * Response to comment on “1-Lipschitz requirement": The OTNN networks analyzed in this paper are associated with the optimal transport loss (refer to line 106 and subsequent lines), necessitating not only a 1-Lipschitz constraint but also a $\nabla_x f=1$ almost everywhere. Properties 1 & 2, proven in this paper, would not be applicable if the Lipschitz constant exceeded 1. Besides, as described in the paper l.239, details on OTNN network architectures and training parameters are given in Appendix A.2.
>
> * Response to comment on “Related Work": We acknowledge in the paper that the link between robustness and explainability has been previously shown, citing three references [11,61,18]. If space permits in the final version, we could include the papers suggested by the reviewer. However, it's worth noting that none of these studies provide a theoretical link between the two elements, which we have proven in the case of OTNN.
> * Response to comment on “Evaluation": Despite length constraints preventing us from including MNIST evaluation in this paper, Figure 1.a illustrates that the gradients are near perfect for this simple dataset. The inclusion of the blockMNIST study would also be hard  due to the paper length limit. However, we carried out experiments on blockMNIST dataset, and we share them with you in the supplementary pdf rebuttal (see general rebuttal section, the first figure for our Results with OTNN, and the second  for a remainder of the cited paper). We hope that it will convince you that gradients of OTNN (due to optimal transport) are even more relevant than those of adversarial trained networks. And as a consequence, human alignment for OTNNs  is more than just a ‘debugging’ feature, but a real novelty.

---

> > ### Comment · Reviewer_WMVT · 2023-08-19
> > **Acknowledging Response**
> >
> > I appreciate the authors' response, and the new updated experiments, it does clear up my concerns.
> >
> > - **Related work**: I agree that the authors cited work showing that there is a link between robustness and explainability. I am familiar with the papers you cited, but I think the three I recommended are more closely related to the phenomenon you've observed here. These three papers are trying to answer the question: "why are the gradients of standard, i.e. unregularized models not sensitive/faithful to the model's output." Here you've shown that the gradients of an OTNN model sidestep these issues. This goes to your point about "none of these studies provide a theoretical link between the two elements". I would be careful making that statement because it is not true. The Shah et. al paper shows analysis that indicates why the gradients of an unregularized model is not sensitive to the model output, but that of an adversarially robust model is. Srinivas et. al. show why we have issues with the logit gradient as well, i.e. reinterpreting it as a relating to a class-conditional generative model. To me, all three papers provide some 'theoretical' link between robustness and explainability. I would not claim that this work is the first to provide such a link. I don't think these related work detract or reduce your work in anyway, it might help to engage with these papers on a deeper level. I think some of the beneficial properties that you noticed for OTNN could be related to the insights from these papers; in my opinion, these three papers are more relevant than what you've cited. In addition, I went through the theorems here again, and you show that indeed the gradient of an OTNN model should have desirable properties, but it is still unclear to me 'why', i.e., what is it about OTNN training or scheme that confers these beneficial properties to their gradients. This is not necessarily a question that you need to answer in this paper, but the three papers I cited are hinting at what an alternative training scheme that confers beneficial properties on a model's gradients might need to have.
> >
> > - **Evaluation**: First, the authors should be careful, a statement like: "gradients of OTNN (due to optimal transport) are even more relevant than those of adversarial trained networks" is not justified. However, I understand what you are alluding to based on the block MNIST example, but there is no justification to make this statement. My conjecture is that OTNN training probably provides a similar benefit as adversarial training does but with the benefit that the model is more accurate than an adversarially trained model. However, my main issue is with the point that because the gradients of the OTNN model align with human attention then they are 'better'. This is not a good justification. What if the underlying model were relying on signals different from those that a human would've used to solve the problem? In this case, it would still be fine for the gradients of the OTNN model to not align with human attention since it is reflecting what the underlying model is doing.
> >
> > Overall, I think the two points above a mostly minor and do not affect the key message of this work. The paper shows some intriguing insights about OTNN models and their gradients.

---

> > > ### Author Response · Authors · 2023-08-20
> > > **Re: Acknowledging Response**
> > >
> > > Thank you for opening this discussion.
> > > We acknowledge that you mention “the two points are mostly minor”, but since this discussion aims to clarify some points, we will take the opportunity to address them..
> > >
> > > * Concerning the literature. We agree that the suggested papers are highly relevant to assess the link between robustness and explainability. We intend to include citations to these works in our final paper and provide insights derived from them.
> > > * Regarding your sentence: “you show that indeed the gradient of an OTNN model should have desirable properties, but it is still unclear to me 'why'”:  this is exactly where is one of the important points of this paper and the theoretical link. OTNN are not only 1-Lipschitz but learnt with a speciifc loss (HKR), originally proposed by Serrurier et al., who demonstrated it as the dual loss of an optimal transport problem.They also proved that the solution $f$ to this loss yields $\nabla_x f=1$ almost everywhere. In this paper, we prove an additional property that  the gradient of this optimal solution points (almost surely) in the direction to the image $\gamma_{\pi(x)}$ according to the transportation plan. As said in l.213, this image $\gamma_{\pi(x)}$ lies in the other distribution (class) and stands as “the closest in average on the pairing process”.  We are convinced that none of the previous papers can assert that their gradients point almost surely in the direction of the closest sample of the other class. Obviously these properties pertain to the loss minimizer, and maybe not completely for the learnt model. Nevertheless, the loss minimization enforces the gradients to adhere to  this property. Fig 1, Fig 5 and appendix A.5 show qualitative examples.
> > > * Evaluation Clarification: You are correct in pointing out that our sentence  "gradients of OTNN (due to optimal transport) are even more relevant than those of adversarial trained networks" was not intended as an assertion within the paper. Instead, it was related to the supplementary blockMNIST experiment that you requested. The term ‘relevant’  was employed to resonate with the Assumption presented in Shah et al paper, specifically the notion of task-relevance. We agree that the provided picture gives only a few samples of clean gradients. To thoroughly  compare  (which was not the initial goal of this rebuttal experiment) an evaluation of the proxy-metric provided in the appendix of the referenced paper would be requisite. We could add such an evaluation in the appendix of the final paper.
> > > * Concerning  the “human alignment”: We acknowledge that the alignment between gradients and human attention alone isn't an automatic indicator of explanation quality. But it comes with numerous other XAI metrics, as well as the supplementary the proposed blockMNIST extra experiments.  Fel et al ‘s work  [19] discussed about superhuman accuracy and exploitation of biases. They argue that, at least on Imagenet,  “there are facets of human perception that are not captured by DNNs”. We see 3 interest on the results: (i) it is a promising results for the neuroscience community, deeply interested in predicting human gaze, and (ii) it suggest reduced reliance on unreliable cues.Finally (iii), we believe that the Alignment of human and machine explanations serves as a credible pathway towards more trustworthy models.

---

> > > > ### Comment · Reviewer_WMVT · 2023-08-21
> > > > **Thanks for the response**
> > > >
> > > > The response has clarified my concerns. Overall, I think this work has nice findings, so I am inclined to maintain my current score.

---

> ### Author Response · Authors · 2023-08-11
> **Quick follow-up**
>
> Thank you for taking the time to review our work. We hope that our rebuttal has addressed your questions and concerns. Please let us know if you have any unresolved concerns or additional questions about the paper or our rebuttal

---

### Official Review · Reviewer_LnPf · 2023-07-08

**Soundness:** 3 good
**Presentation:** 3 good
**Contribution:** 2 fair
**Rating:** 5
**Confidence:** 4

**Summary:**

The paper presents a theoretical analysis of 1-Lipschitz constrained neural networks that are trained based on optimal transport dual loss for image classification task. The optimal classifier f(.) is trained by minimizing the hinge regularized KR optimization loss. It's been shown that the gradient of optimal solution is aligned with the transportation plan direction and the L1 norm of it at x is a certifiable lower bound of classifier robustness. So the close decision boundary to x is also along with the gradient direction at distance of the lower bound which is |f*(x)|.

The experimental results show that such a model performs better than its unconstrained counterpart in XAI metrics like \mu fidelity and stability spearman rank on multiple datasets including FashionMNIST, CelebA, CatvsDog and Imagenet. It's been shown that OTNN models  explanations are more aligned with human attention that makes the model performance more predictable.

**Strengths:**

- The paper provides a theoretical analysis of properties of 1-Lipschitz constrained neural networks. I consider this type of analysis highly valuable to the field!

- The authors provide empirical study about the effectiveness of OTNNs on improving XAI metrics. Considering multiple datasets, it's been shown that the effectiveness is scalable to larger dataset and larger networks.

- Several properties of OTNN gradient has been investigated and shown that, their saliency maps are aligned with human attention that improves the predictability of network behavior. They also draw connection between gradient and counterfactual examples and adversarial attacks to OTNNs.



**Weaknesses:**

- While I appreciate the theoretical analysis of OTNNs, they are mostly compilation of analysis in [53] and [26] and [6].

- There is no ablation study on activation functions and normalization methods besides GroupSort and Spectral Norm even on a single dataset. It would be nice to evaluate the empirical effect of network depth on XAI metrics.



**Questions:**

- It's been proven that 1-Lipschitz neural network are as accurate as unconstrained neural networks [ref.1 ], why are the OTNN models are practically less accurate.

- Figure 2 demonstrates the level set of an OTNN binary classifier, how does the unconstrained network decision boundary look like? I appreciate if you could provide the such figure.

- Could you provide the table corresponding to the Figure 4 that clarifies what each of the point in the figure is and what is the exact value of accuracy and human feature alignment in addition to the details of experimental setup that leads to the figure?



[ref.1] Béthune, Louis, et al. "Pay attention to your loss: understanding misconceptions about lipschitz neural networks." Advances in Neural Information Processing Systems 35 (2022): 20077-20091.

**Limitations:**

There is no negative societal impact of this paper as it is a theoretical paper.

---

> ### Author Rebuttal · Authors · 2023-08-03
>
> We greatly appreciate your thorough and insightful review, and have taken your valuable feedback into consideration.We are delighted that you mention that “this type of analysis  [is] highly valuable to the field!”
> Please find below answers to your main questions:
> * Response to Weaknesses:
>    * In fact, properties 1 and 2 are original contributions, not found or proven in prior literature. While the link between robustness and explainability may not be new, our work with the OTNN framework establishes explicitly and theoretically this link, marking a significant novelty.
>   * Furthermore, This paper demonstrates the first results based on OTNN on large scale networks and datasets. This was achieved by slightly modifying the multiclass loss function, resulting in enhanced accuracy.
>    * Regarding the ablation study, the effects of the different 1-Lipschitz layers have been thoroughly investigated for instance  in [48] and [4]. Since OTNN is based on optimal transport, which requires $\nabla_x f=1$ almost everywhere, we opted for layers that best fit these requirements. Although an ablation study might have value from an XAI perspective, it is beyond the theoretical scope of this paper.
> * Response to Questions:
>    * Despite the fact that 1-Lipschitz networks can be as accurate as unconstrained ones, OTNN and the optimal transport loss impose stronger guarantees than a mere 1-Lipschitz constraint. Notably, it's recognized that robustness guarantees, and by consequence the induced XAI guarantee in our case, influence accuracy. This impact remains negligible for all studied datasets except for  ImageNet (see lines 240-246). For ImageNet, the accuracy of OTNN aligns with several classical networks (see Fig. 4), and provides substantial XAI properties. For a simpler and  fairer comparison, we utilized a standard ResNet architecture, but it remains an open question whether standard architectures are the most suitable for 1-Lipschitz networks (and thus for  OTNN).
>    * The intent of Figure 2 was to demonstrate the unique characteristics of OTNN  and an understandable illustration of Propositions 1 and 2. The level sets of unconstrained networks can be challenging to interpret, given that there are no certificates for the output (output values and gradient norm can be unpredictably high). All our attemps to generate such a figure with unconstrained networks lead to shifted points  $x-f(x)\nabla_x(f(x))$  far outside the two Koch snowflakes and thus unreadable.
>   * We will incorporate a table with precise values and references of networks of Fig 4 in the final version and  develop the discussion in the main text. The experimental setup strictly followed the methodologies of [35] and [17], and their code was used for our experimentation. We will amend line 278 to make this aspect clearer.

---

> ### Author Response · Authors · 2023-08-11
> **Quick follow-up**
>
> Thank you for taking the time to review our work. We hope that our rebuttal has addressed your questions and concerns. Please let us know if you have any unresolved concerns or additional questions about the paper or our rebuttal

---

### Author Rebuttal · Authors · 2023-08-03

Dear Reviewers,
We extend our gratitude to the reviewers for dedicating their time to thoroughly read and assess our paper. Their critiques have been sharp and insightful, and we appreciate that all four reviewers recommend accepting the paper. Since there is still a shared concern regarding the need to clarify the novelty of this work, which might not have been fully emphasized in the initial submission, we want to address it globally here.

* Key novelty lies in the **theoretical link** established between orthogonality, 1-Lipschitz constraint, robustness, and explainability within the context of optimal transport. This unique link is **not only due to 1-Lipschitz constraint but also to the chosen loss, based on optimal transport, and $\nabla_x f=1$ almost everywhere properties** imposed on the layers. This theoretical link is proven through **Proposition 1 and 2 which are original contributions**. These allow our proposed framework, OTNN, to offer both accuracy and robustness, and more importantly, an enhanced level of explainability, which is a key requirement in critical domains like healthcare and transportation.
* **First certifiable 1-Lipschitz Neural Network on ImageNet dataset**: We have also demonstrated that our approach, with a slightly modified multiclass Optimal Transport  loss, can be applied to complex tasks as Imagenet with theoretical guarantees. It maintains competitive accuracy (70% with a Resnet50 like architecture) and possesses remarkable explainability properties that surpass those of state-of-the-art models, even at iso-accuracy.
* We agree with your comments on the large literature about the link between robustness and xai, and will reinforce this part in the final paper (3 citations are already in the paper). But  our approach has shown a **clear advantage over previously published methods, offering theoretical guarantees both on robustness and explainability**, and demonstrating increased alignment with human interpretation.

We trust that these enhancements and clarifications to our paper underscore its importance and significance in the realm of explainable AI. We hope that you will reconsider the scoring in light of these arguments.

Moreover, we have diligently incorporated the necessary citations and clarifications into the appropriate sections. We will also address each reviewer’s remaining comments directly.

Thank you once again for your time and thoughtful critique.

---

### Decision · Program_Chairs · 2023-09-21

**Decision:**

Accept (poster)

**Comment:**

All reviewers acknowledge the importance of both the novel theoretical results as well as the experiments on large datasets and complex tasks. The AC does not find reason to overturn this unanimous suggestion and recommends acceptance. However, as noted by most reviewers, it is crucial to cite and properly discuss the works that try to make an empirical, theoretical, or formal connection between bounded or 1-Lipschitz and robustness and explainability. Quite a few are missing and occasionally statements of originality/novelty are too bold. The camera-ready version is expected to be revised accordingly.